# N-AS-triggered SPMs are direct regulators of microglia in a model of Alzheimer's disease

Ju Youn Lee[1,2,3,10], Seung Hoon Han[1,2,3,10], Min Hee Park[1,2,3], Im-Sook Song[4], Min-Koo Choi[5], Eunsoo Yu [6], Cheol-Min Park[6], Hee-Jin Kim [7], Seung Hyun Kim [7], Edward H. Schuchman[8], Hee Kyung Jin[1,9 ✉] & Jae-sung Bae[1,2,3 ✉]

Sphingosine kinase1 (SphK1) is an acetyl-CoA dependent acetyltransferase which acts on cyclooxygenase2 (COX2) in neurons in a model of Alzheimer's disease (AD). However, the mechanism underlying this activity was unexplored. Here we show that N-acetyl sphingosine (N-AS) is first generated by acetyl-CoA and sphingosine through SphK1. N-AS then acetylates serine 565 (S565) of COX2, and the N-AS-acetylated COX2 induces the production of specialized pro-resolving mediators (SPMs). In a mouse model of AD, microglia show a reduction in N-AS generation, leading to decreased acetyl-S565 COX2 and SPM production. Treatment with N-AS increases acetylated COX2 and N-AS-triggered SPMs in microglia of AD mice, leading to resolution of neuroinflammation, an increase in microglial phagocytosis, and improved memory. Taken together, these results identify a role of N-AS in the dysfunction of microglia in AD.

[1] KNU Alzheimer's Disease Research Institute, Kyungpook National University, Daegu 41566, South Korea. [2] Department of Physiology, Cell and Matrix Research Institute, School of Medicine, Kyungpook National University, Daegu 41944, South Korea. [3] Department of Biomedical Science, BK21 Plus KNU Biomedical Convergence Program, Kyungpook National University, Daegu 41944, South Korea. [4] College of Pharmacy and Research Institute of Pharmaceutical Sciences, Kyungpook National University, Daegu 41566, South Korea. [5] College of Pharmacy, Dankook University, Cheon-an 31116, South Korea. [6] Department of Chemistry, Ulsan National Institute of Science and Technology (UNIST), Ulsan 44919, South Korea. [7] Department of Neurology, Hanyang University College of Medicine, Seoul 04763, South Korea. [8] Department of Genetics and Genomic Sciences, Icahn School of Medicine at Mount Sinai, New York, NY 10029, USA. [9] Department of Laboratory Animal Medicine, College of Veterinary Medicine, Kyungpook National University, Daegu 41566, South Korea. [10] These authors contributed equally: Ju Youn Lee, Seung Hoon Han. ✉email: hkjin@knu.ac.kr; jsbae@knu.ac.kr

Cyclooxygenase-2 (COX2), a major factor in inflammatory reactions[1], catalyzes the conversion of its substrates, such as arachidonic acid (AA), eicosapentaenoic acid (EPA), and docosahexaenoic acid (DHA), to pro-inflammatory lipid mediators including prostaglandins (PGs), 11-hydroxyeicosatetraenoic acid (HETE)[2], 11-hydroxyeicosapentaenoic acid or 15-hydroxyeicosapentaenoic acid (HEPE)[3], and 13-hydroxydocosahexaenoic acid (HDHA)[4,5]. Interestingly, when COX2 is acetylated by acetylsalicylic acid (ASA, aspirin), the acetylated COX2 switches its catalytic activity to convert AA, EPA, and DHA to 15-HETE, 18-HEPE, and 17-HDHA, respectively[2–5]. These molecules can be subsequently converted to 15R-lipoxin A4 (15R-LXA$_4$) and resolvins (Rv) E1, D1, D2, and D3, known as specialized pro-resolving mediators (SPMs), in the presence of 5-lipoxygenase (5-LOX)[6,7]. SPMs have potent pro-resolving actions, leading to cessation of immune cell infiltration, downregulation of pro-inflammatory and upregulation of anti-inflammatory mediators, and promotion of phagocytosis and tissue regeneration[6,7]. Also, recent studies have reported that SPMs are reduced in neuroinflammatory diseases, such as Alzheimer's disease (AD), resulting in dysfunction of microglia[8,9]. However, the mechanisms that underlie the regulation of microglia via SPMs have not been identified in most neuroinflammatory diseases, including AD.

Sphingosine kinase-1 (SphK1) is an ATP-dependent lipid kinase that catalyzes the conversion of sphingosine to sphingosine-1-phosphate (S1P)[10]. Recently, we demonstrated a role for SphK1 as an acetyl-CoA dependent cytoplasmic acetyltransferase with activity towards COX2[11]. We also confirmed that neuronal SphK1 levels are reduced in AD brain, and that increased SphK1 promoted SPM secretion in neurons, especially 15R-LXA$_4$, by acetylating serine residue 565 (S565) of COX2, resulting in improvement of AD-like pathology. However, the biochemical and specific AD pathogenic mechanisms underlying SphK1-mediated COX2 acetylation is not fully understood.

Here, we show that acetyl-CoA binds the ATP binding site in SphK1, and that reaction of acetyl-CoA and sphingosine within SphK1 generates N-acetyl sphingosine (N-AS). N-AS produced by SphK1 binds COX2, and acetylates the S565, which corresponds to the residue we previously showed was acetylated by SphK1[11]. N-AS-acetylated COX2 increases production of 15-HETE, 18-HEPE, and 17-HDHA, which can be subsequently converted to SPMs, and enhances SPM production, including 15R-LXA$_4$, RvE1, and RvD1, similar to aspirin-acetylated COX2[2–7]. Furthermore, N-AS generation was decreased in microglia of AD, leading to reduction of acetyl-S565 COX2 and N-AS-triggered SPMs. These findings also led us to investigate the role of N-AS in AD microglia. The treatment of APP/PS1 mice with N-AS increased N-AS-induced SPMs in microglia via S565 acetylation of COX2, resulting in the direct regulation of microglial function. Finally, the N-AS-triggered SPMs led to resolution of neuroinflammation and upregulation of several reactive microglial genes linked to phagocytosis compared with untreated APP/PS1 microglia, leading to amelioration of AD pathology. In addition, amyloid β (Aβ)-treated human microglia also showed reduction of N-AS generation, and N-AS treatment of these human cells improved SPM production and phagocytosis capacity as well. Overall, these results reveal a biosynthetic mechanism and function of N-AS, which leads to S565 acetylation of COX2 and production of SPMs. They also reveal the relation of N-AS with microglial regulation in AD pathogenesis, and suggest a potential therapy for neuroinflammatory diseases, such as AD, using N-AS or related derivatives that could be evaluated in the future.

## Results

**Residue S565 of COX2 is acetylated by N-AS.** To further study the role of N-AS as an intermediate in SphK1-mediated COX2 acetylation, we performed a binding assay of N-AS to COX2. Incorporation of N-AS to COX2 became saturated with increased concentrations of N-AS, yielding $K_M$ and $K_{cat}$ values of 35.48 μM and 0.51 min$^{-1}$, respectively (Fig. 1a). We also measured the acetylation level of COX2 in the presence of [$^{14}$C] N-AS. N-AS-mediated COX2 acetylation displayed higher-level acetylation compared with aspirin or SphK1-mediated COX2 acetylation (Fig. 1b). These results indicated that N-AS, as an intermediate for SphK1-mediated COX2 acetylation, had high binding affinity and induced high-level acetylation of COX2.

Next, to identify which site on COX2 was acetylated by N-AS, we treated N-AS with COX2. From these studies we found that S565 on peptide 565-GCPFTSFSVPDPELIK-575 of COX2 was acetylated in the presence of N-AS, the same site that we found acetylated by SphK1 in our previous study[11] (Fig. 1c, d). To further characterize acetylated S565 of COX2 by N-AS, we generated rabbit polyclonal and mouse monoclonal antibodies, ac-S565. Incubation of COX2 with N-AS led to strong ac-S565-positive signals in western blot analyses compared with naïve COX2 and aspirin-treated COX2 (Fig. 1e). Projection of the acetylation site on the theoretical crystal structure of human COX2 showed that S565 was located on the catalytic domain of the enzyme[12] and in the vicinity of asparagine 181 (N181), threonine 564 (T564), and serine 567 (S567) (Fig. 1f). Based on the theoretical crystallographic model, we propose a possible mechanism of N-AS-acetylated COX2 in which acetyl transfer to S565 in the COX2 active site via N-AS was promoted by the hydrogen bonding of N181 and T564 with the acetyl group. Additional hydrogen bonding of S567 with the C3 hydroxyl group appeared to further increase the binding affinity of N-AS (Fig. 1g). To establish the validity of this mechanism, we mutated residues S565, N181, T564, and S567 of COX2 and performed kinetic and acetylation analysis. The binding affinity and acetylation of COX2 were significantly decreased in the mutant COX2, including S565A, N181A, T564A, and S567A, compared with COX2 WT. Interestingly, S565A showed the lowest binding affinity and acetylation level compared to the other mutants, suggesting that S565 is the main target site of N-AS-mediated COX2 acetylation (Fig. 1a, h).

**N-AS induces production of SPMs by COX2 acetylation.** To investigate the production of SPMs by N-AS-acetylated COX2, we performed COX2 product profiling for AA, EPA, and DHA using recombinant COX2 treated with N-AS in the presence of 5-LOX (Supplementary Table 1). Consistent with previous studies, including ours[2–7,11], N-AS-treated COX2 significantly increased 15-HETE, 18-HEPE, and 17-HDHA, known as SPM precursors[6,7], compared with naïve COX2 and aspirin-treated COX2 (Fig. 2a and Supplementary Table 1). Moreover, the N-AS treated COX2 reduced PGs, such as PGH$_2$ (Supplementary Table 1). Next, several SPMs were identified using LC-MS/MS. Interestingly, SPMs, including 15R-LXA$_4$, RvE1, and RvD1, were produced by N-AS-acetylated COX2 (Fig. 2b). The level of SPMs induced by N-AS exhibited a marked increase compared with those produced by COX2 without treatment, and was similar to those produced by aspirin (Fig. 2c). To gain more direct insights into the relationship of N-AS-acetylated COX2 and SPM production, we verified the production of SPMs in the COX2 S565A mutant. The S565A mutant led to considerably decreased production of SPMs after treatment of N-AS,

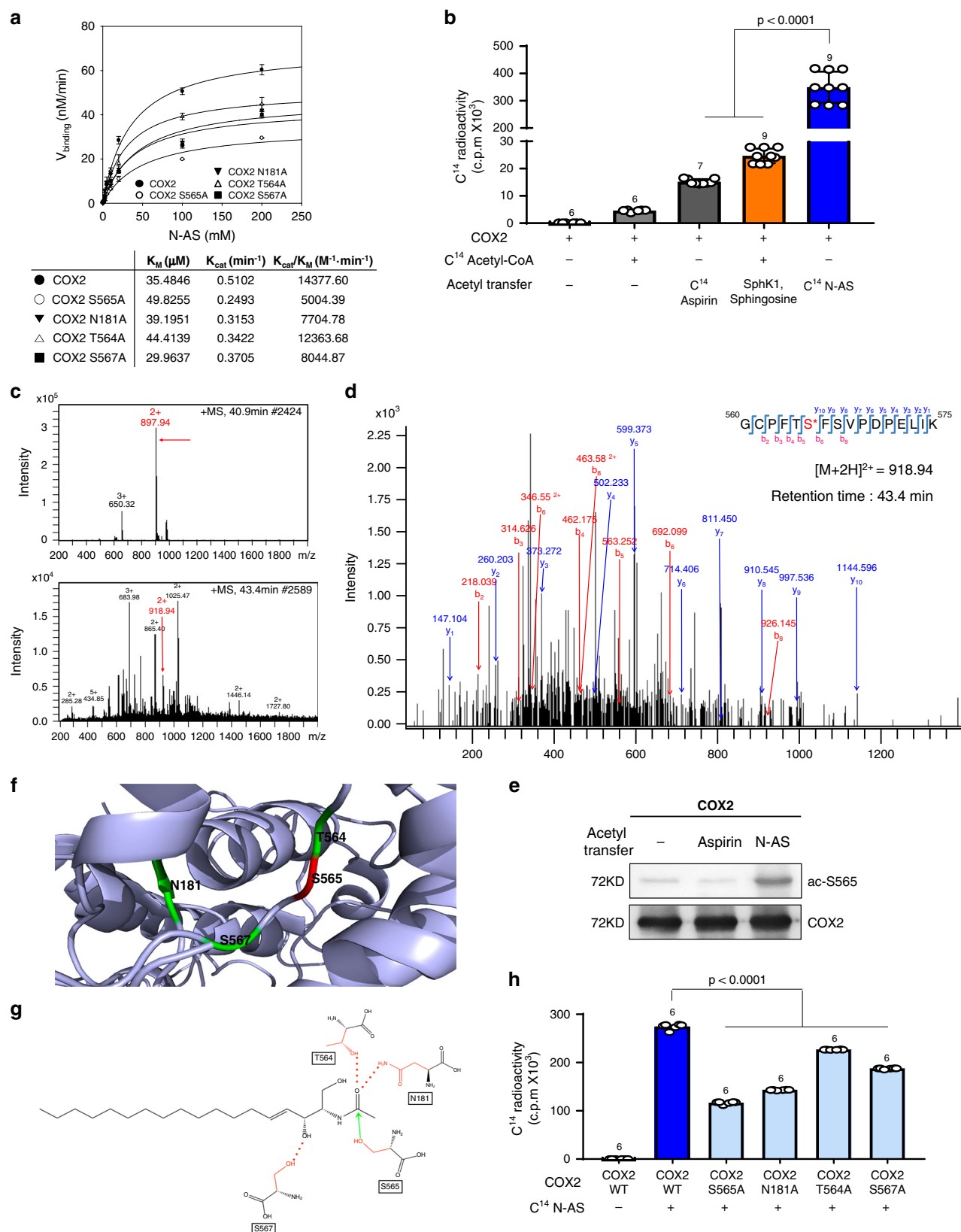

although there were no differences after treatment with aspirin, indicating that S565 in N-AS-acetylated COX2 is important for production of SPMs (Fig. 2c). Overall, we confirmed that N-AS-acetylated COX2 produced SPMs, suggesting the potential pathological role and therapeutic use of N-AS in inflammatory diseases, including AD.

**Aβ induce loss of N-AS generation in neuron and microglia.**
Previously, SphK1-mediated COX2 acetylation was observed in neurons, and was found to be decreased in neurons derived from AD[11], suggesting that N-AS produced by SphK1 might be important for central nervous system (CNS) cells. To investigate the generation and roles of N-AS in CNS cells, primary culture of

**Fig. 1 N-AS acetylates S565 in COX2. a** N-AS binding activity of COX2 WT, COX2 S565A, COX2 N181A, COX2 T564A, and COX2 S567A was analyzed by filter binding assay. The binding velocity ($V_{binding}$) of [$^{14}$C] N-AS to COX2 was plotted to the N-AS concentration and the nonlinear regression analysis of the saturated plot yielded the kinetic parameters such as $K_{cat}$ (catalytic constant) and $K_M$ (Michaelis–Menten constant) for N-AS and COX2 binding activity ($n = 3$ independent experiments per group). **b** Acetylation assay of purified COX2 protein treated with [$^{14}$C] N-AS. The purified COX2 protein incubated in the presence of [$^{14}$C] N-AS for 2 h at 37 °C and then COX2 was analyzed on scintillation counter. [$^{14}$C] aspirin-treated COX2 protein and [$^{14}$C] acetyl-CoA treated COX2 protein with SphK1 and sphingosine were positive control ($n = 6$–9 per group). **c** LC-MS spectra of peptide 560-GCPFTSFSVPDPELIK-575 ($m/z = 918.94$) of COX2 acetylated by N-AS. **d** LC-MS/MS spectra of ac-S565 in 560-GCPFTSFSVPDPELIK-575 of COX2. **e** Recombinant COX2 protein was incubated in the presence or absence of 2 mM N-AS or aspirin as indicated and S565 acetylation was determined by western blotting using anti-ac-S565 and COX2 antibodies. Data were replicated in six independent experiments with similar results. **f** The theoretical crystallographic model of human COX2 structure was taken from Protein Data Bank file 1V0X. S565 and residues (N181, T564, and S567) involved in the catalysis of COX2 were projected on the crystal structure of COX2 using PyMOL. Residues are shown as well: (1) S565 (red); (2) N181, T564, and S567 (green). **g** Proposed mechanism for the acetylation of S565 of COX2 by N-AS. **h** Acetylation assay of COX2 WT, COX2 S565A, COX2 N181A, COX2 T564A, and COX2 S567A treated with [$^{14}$C] N-AS. ($n = 6$ independent experiments per group). **b**, **h** One-way analysis of variance, Tukey's post hoc test. All error bars indicate s.e.m. Source data are provided as a Source data file.

neuron, microglia and astrocyte was prepared from C57BL/6 mice, and confirmed generation of N-AS in these cells. N-AS was detected in neurons and microglia, but not in astrocytes (Fig. 3a). Also, to investigate whether N-AS was correlated with the pathogenesis of AD, we treated cells with 10 μM Aβ. The treatment with Aβ reduced N-AS generation in neurons and microglia, indicating the possible relationship of N-AS with neurons and microglia in AD (Fig. 3a). Next, we assessed N-AS-acetylated COX2 in these cells using the ac-S565 antibodies. Consistent with N-AS generation, N-AS-mediated acetyl-S565 COX2 was observed in both neurons and microglia, but not in astrocytes. Further, these studies confirmed that in the presence of Aβ, the levels of N-AS-acetylated COX2 were decreased in both neurons and microglia (Fig. 3b, c). These results indicated that N-AS was generated, and acetylated S565 of COX2 in neuron and microglia, which was abrogated with Aβ treatment. They also indicated that astrocytes were not involved in this mechanism.

To further determine why the generation of N-AS was reduced in neurons and microglia treated with Aβ, we examined factors related to the generation of N-AS, including SphK activity, acetyl-CoA, and sphingosine. SphK activity was decreased in Aβ-treated neurons compared with neurons without treatment, although there was no significant difference in microglia, similar to our previous study[11] (Fig. 3d). By contrast, the levels of acetyl-CoA were significantly reduced in Aβ-treated microglia compared with microglia without treatment, but not in Aβ-treated neurons (Fig. 3e). Sphingosine levels were unchanged by Aβ in these cells (Fig. 3f). Interestingly, supplementation with SphK1 in Aβ-affected neurons enhanced N-AS generation, but not acetyl-CoA and sphingosine. In contrast, in Aβ-affected microglia, the supplementation with acetyl-CoA improved N-AS generation, but not SphK1 and sphingosine (Fig. 3g). In addition, S565 acetylation of COX2 and SPM expression, which were decreased in Aβ-treated neurons and microglia, improved after treatment with SphK1 and acetyl-CoA in Aβ-affected neurons and microglia, respectively, similar to restoration by treatment of N-AS (Fig. 3h, i). These data showed that Aβ led to reduction of N-AS generation through the loss of SphK activity in neurons and the deficiency of acetyl-CoA in microglia, indicating different mechanisms for these two important cell types. Overall, our results indicated that N-AS was generated, followed by S565 acetylation of COX2 and SPM expression, in neurons and microglia of the CNS. This pathway was disrupted by different mechanisms in these two cell types upon exposure to Aβ. Notably, the levels of N-AS, acetyl-S565 COX2, and SPMs expression in microglia were higher than those in neurons, suggesting the generation and role of N-AS was more essential in microglia than neurons. Finally, these results suggested that the reduced generation of N-AS might influence disease progression and/or pathogenesis in AD microglia, and might be treated by N-AS supplementation.

**Generation and function of N-AS is reduced in AD microglia.** Next, to assess the generation and roles of N-AS in AD pathogenesis, we first determined N-AS generation in microglia and neurons derived from WT mice and APP/PS1 mice. N-AS was detected in microglia and neurons derived from WT mice, and decreased in those derived from APP/PS1 mice compared with WT mice (Fig. 4b). Specificity of ac-S565 COX2 immunoreactivity was also observed in microglia and neurons of WT mice, and significantly reduced in microglia and neurons of APP/PS1 mice compared with WT mice (Fig. 4c, d and Supplementary Fig. 4a). Consistent with CNS cell studies above, the levels of N-AS and acetyl-S565 COX2 were higher in microglia than in neurons, suggesting N-AS-acetylated COX2 was more active in microglia than in neuron. These results indicated that N-AS might play important roles in microglia of AD. In addition, to assess the influence of N-AS in AD pathogenesis, we injected N-AS daily subcutaneously (s.c.) for 8 weeks to 7-months-old APP/PS1 mice until the age of 9 months (Fig. 4a). In APP/PS1 mice treated with N-AS, the levels of N-AS and N-AS-mediated COX2 acetylation were more restored in microglia of APP/PS1 mice compared with those in neurons (Fig. 4b–d and Supplementary Fig. 4a). These data indicated that microglia were the likely target cells responsible for AD pathogenesis via N-AS.

To further investigate why AD pathology was more dependent on N-AS in microglia compared to neurons, we first assessed expression of COX2 in microglia and neurons. COX2 expression was significantly increased in APP/PS1 microglia compared with WT microglia beginning at 5 months of age. In contrast to the microglia, no significant differences in COX2 expression was observed in neurons (Supplementary Fig. 4b, c). Next, to confirm whether N-AS was bound to and acetylated COX2 in the APP/PS1 microglia and neurons, we isolated microglia and neurons after [$^{14}$C] N-AS administration to WT and APP/PS1 mice, and checked [$^{14}$C] radioreactivity of COX2 purified from these cells. We found that [$^{14}$C] N-AS was more avidly bound to and acetylated COX2 of APP/PS1 microglia compared with COX2 in WT microglia; no significant differences were observed in neurons (Supplementary Fig. 4d). These results indicated that the high COX2 expression of microglia might be prone to increased binding and acetylation by N-AS compared with that in neurons in APP/PS1 mice.

Finally, to assess the ability of N-AS-acetylated COX2 to produce SPMs in microglia of APP/PS1 mice, we performed lipidomic profiling analysis in microglia derived from WT, APP/PS1, and APP/PS1 mice treated with N-AS (Supplementary Table 2). The SPM precursors, including 15-HETE, 18-HEPE, and 17-HDHA, were decreased in APP/PS1 microglia versus WT microglia. However, microglia derived from APP/PS1 mice treated with N-AS had levels that were restored to those of WT

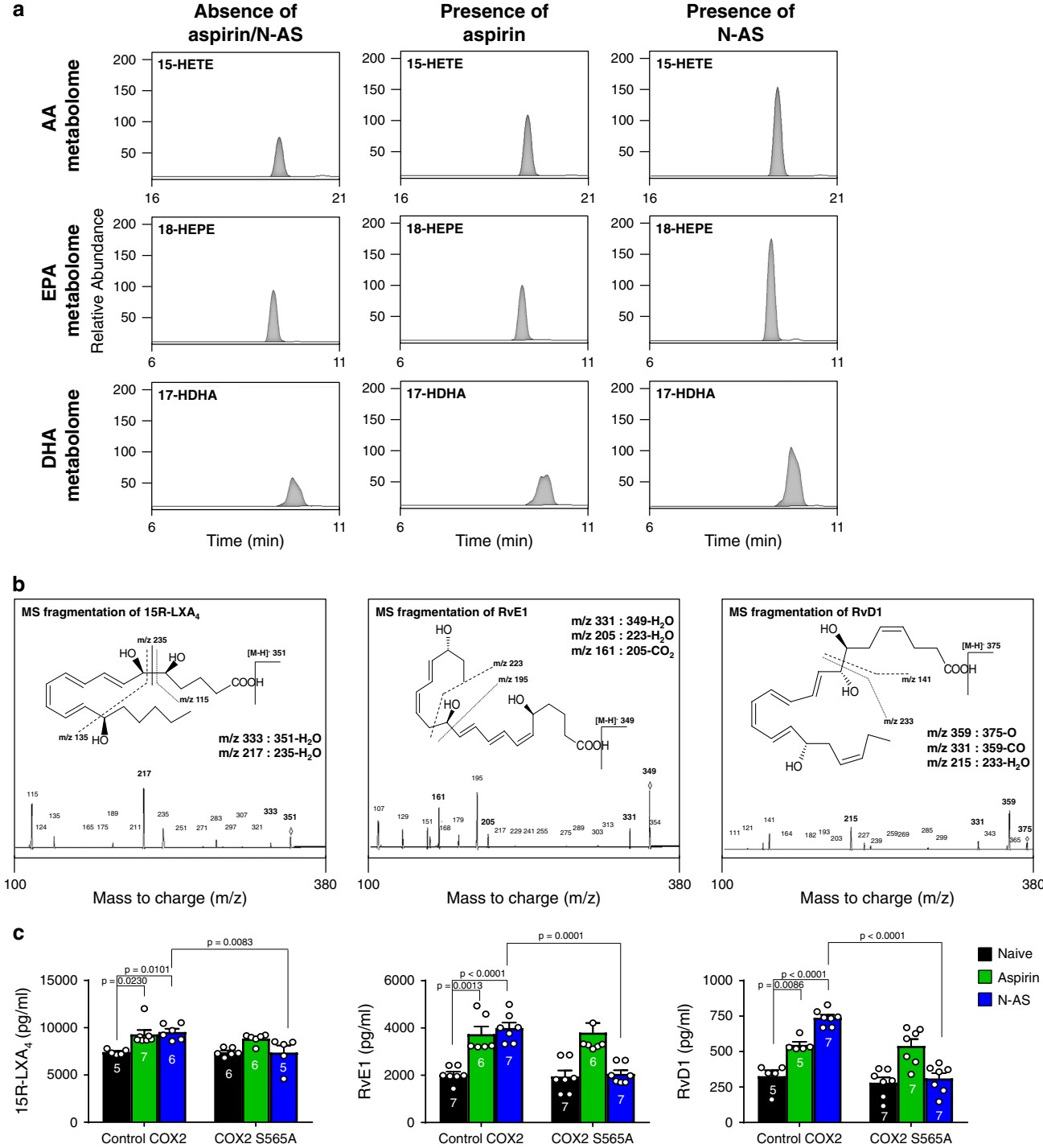

**Fig. 2 N-AS-acetylated COX2 produces SPMs. a** Human recombinant COX2 treated in the presence or absence of 500 µM N-AS or aspirin was incubated with AA, EPA, or DHA in presence of human 5-LOX, and then SPM precursors were identified using systematic LC-MS/MS. Representative chromatogram showing the SPM precursors (Top, 15-HETE; Middle, 18-HEPE; Bottom, 17-HDHA). **b** Human recombinant COX2 treated in the presence of 500 µM N-AS or aspirin was incubated with AA, EPA, or DHA in presence of human 5-LOX. Related MS/MS spectra employed for identification of SPMs. 15R-LXA$_4$, RvE1, and RvD1. **c** Quantification of 15R-LXA$_4$, RvE1, and RvD1 in COX2 WT and COX2 S565A treated with 5-LOX in presence of N-AS, aspirin or not ($n = 5$–7 independent experiments per group). **c** One-way analysis of variance, Tukey's post hoc test. All error bars indicate s.e.m. Source data are provided as a Source data file.

microglia (Supplementary Fig. 4e; Supplementary Table 2). Consistent with these observations, the SPMs themselves, such as 15R-LXA$_4$, RvE1, and RvD1, exhibited a marked decrease in microglia derived from APP/PS1 mice compared with microglia derived from WT mice. These levels were restored in microglia derived from APP/PS1 mice treated with N-AS (Supplementary Fig. 4f, g; Supplementary Table 2). Taken together, these results demonstrated that N-AS generation was mainly decreased in microglia of APP/PS1 mice, which resulted in reduced acetyl-S565 COX2 and SPM expression, and N-AS treatment led to potent anti-inflammatory actions by inducing N-AS-triggered SPMs via S565 acetylation of COX2.

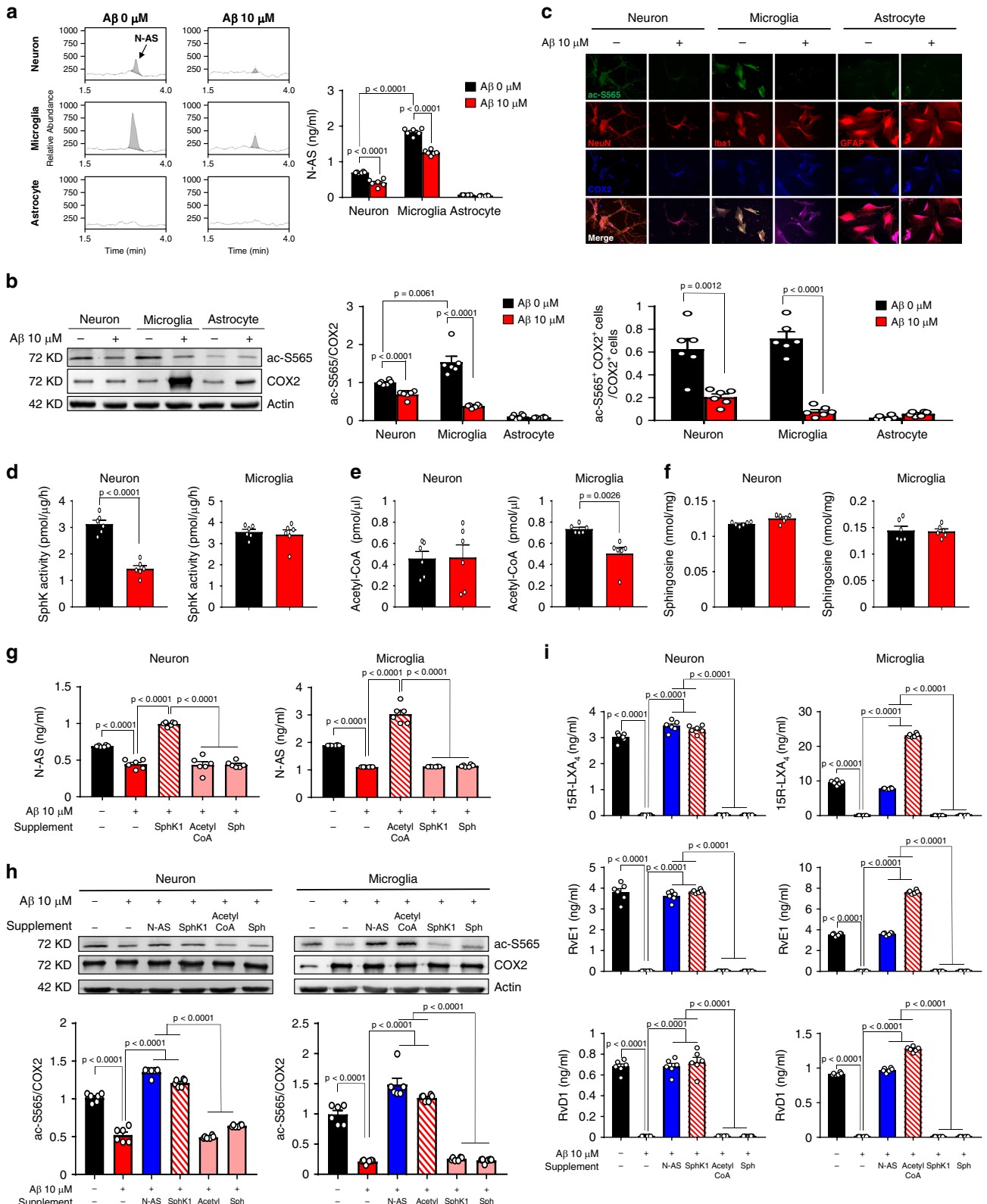

**N-AS-triggered SPMs regulate microglial characterization.** To investigate the direct effect of N-AS-triggered SPMs on microglial regulation, we first examined microglial activation by determining the activities of pro-inflammatory (CD86)[13] or anti-inflammatory (CD206)[13] molecules, in WT, APP/PS1, and N-AS-treated APP/PS1 mice. The CD206−CD86+ pro-inflammatory microglia were reduced, and the CD206+CD86− anti-inflammatory microglia

were abundant in APP/PS1 mice treated with N-AS compared with APP/PS1 mice treated with vehicle, indicating that N-AS-induced SPMs improved the anti-inflammatory activity of microglia in APP/PS1 mice (Fig. 4e).

Next, to further examine the regulation of microglia by N-AS-triggered SPMs, we analyzed the gene expression patterns of microglia derived from WT, APP/PS1, and N-AS-injected

**Fig. 3 Loss of N-AS generation by Aβ reduces COX2 acetylation and SPMs production in neurons and microglia, except astrocyte. a** The primary culture of neuron, microglia, and astrocyte was prepared from C57BL/6 mice, and N-AS were detected by LC-MS/MS in these cells. Representative chromatograms of N-AS and quantification in neuron, microglia, and astrocyte treated 10 μM Aβ or not (n = 6 per group). **b** Western blotting for ac-S565 and total COX2 in neuron, microglia, and astrocyte treated 10 μM Aβ or not (n = 6 per group). **c** Immunofluorescence images and quantification of neuron (NeuN, red), microglia (Iba1, red), or astrocyte (GFAP, red) with ac-S565 (green) and COX2 (blue) (n = 6 per group, scale bars, 50 μm). **d** SphK activity in neuron and microglia treated 10 μM Aβ or not (n = 6 per group). **e** Analysis of acetyl-CoA in neuron and microglia treated 10 μM Aβ or not using assay kit (n = 6 per group). **f** Detection of sphingosine in neuron and microglia treated 10 μM Aβ or not using UPLC (n = 6 per group). **g** Quantification of N-AS by LC-MS/MS in neuron and microglia treated 10 μM Aβ or not in presence of acetyl-CoA, sphingosine, or SphK1 each (n = 6 per group). **h** Western blotting for ac-S565 in microglia and neuron treated 10 μM Aβ or not in presence of N-AS, acetyl-CoA, sphingosine, or SphK1 each (n = 6 per group). **i** Quantification of 15R-LXA$_4$, RvE1, and RvD1 using systematic LC-MS/MS in microglia and neuron treated 10 μM Aβ or not in presence of N-AS, acetyl-CoA, sphingosine, or SphK1 each (n = 6 per group). **a–f** Student's t-test. **g–i** One-way analysis of variance, Tukey's post hoc test. All error bars indicate s.e.m. Source data are provided as a Source data file.

APP/PS1 mice using RNA seq. Comparative RNA seq analysis revealed that APP/PS1 microglia expressed high levels of genes that induced an inflammatory response (e.g., *Spp1, Itgax, Axl, Lilrb4a, Clec7a, Ccl2, Csf1,* and *Apoe*) and suppressed homeostatic genes (e.g., *Fa2h, Cidea, Neurod1, Gck, Epor,* and *Atp7b*) compared with WT microglia (with <2 Z-score and P < 0.05). These results confirmed the characteristics of the microglial neurodegenerative phenotype of APP/PS1 mice determined in previous studies[14,15]. Interestingly, microglia derived from APP/PS1 mice treated with N-AS showed upregulation of several reactive microglial genes (e.g., *Rap1gap, Tub, Mex3b, Appl2, Adam8,* and *Megf10*) linked to phagocytosis and downregulation of inflammatory and immune molecules (e.g., *Spp1, Itgax, Axl, Lilrb4a, Clec7a, Ccl2, Csf1,* and *Apoe*) compared with microglia derived from APP/PS1 treated with vehicle, indicating improvement of the microglial neurodegenerative phenotype of APP/PS1 mice (Fig. 4f). Furthermore, functional categorization using the DAVID bioinformatics database for GO term enrichment analysis (biological process and molecular function) demonstrated that the differentially expressed genes in microglia of N-AS-treated APP/PS1 mice compared with APP/PS1 mice were strongly associated with an inflammatory response, immune response, and phagocytosis (Fig. 4g, h). KEGG pathway analysis identified a considerable enrichment of genes involved in phagocytosis (e.g., lysosome and phagosome) and immune response (e.g., cytokine–cytokine receptor interaction and chemokine signaling pathway) (Fig. 4i). These results indicated that N-AS-triggered SPMs activated an anti-inflammatory, positive immune response, and enhanced the phagocytic abilities of microglia in N-AS-treated APP/PS1 mice, leading to resolution of neuroinflammation and upregulation of phagocytic microglia in this AD animal model.

**SPMs regulate microglial phagocytosis and immune response**. Based on RNA seq results, we next examined whether the enhancement of phagocytic microglia resulted from N-AS-triggered SPMs and promoted Aβ phagocytosis in N-AS-treated APP/PS1 mice. First, to determine whether N-AS-triggered SPMs restored microglial recruitment to Aβ, we quantified the number of microglia surrounding the plaques, and found that microglial recruitment was increased in APP/PS1 mice treated with N-AS compared to APP/PS1 mice treated with vehicle (Fig. 5a). We next performed a phagocytosis assay using live brain slices of WT, APP/PS1 mice injected with vehicle and APP/PS1 injected with N-AS mice. More phagocytic microglia were found in APP/PS1 mice treated with N-AS compared to APP/PS1 mice treated with vehicle, using a fluorescent bead phagocytosis assay (Fig. 5b). To further investigate this effect, the Aβ phagocytic activity of microglia was evaluated in APP/PS1 mice treated with vehicle and APP/PS1 treated with N-AS mice. APP/PS1 mice treated with N-AS showed demonstrably increased number of microglia containing co-stained lysosomes and Aβ compared to APP/PS1 mice treated with vehicle

(Supplementary Fig. 5a). Also, we performed morphological characterization of microglia surrounding Aβ in APP/PS1 mice treated with vehicle and APP/PS1 treated with N-AS mice, and found that the phagocytic morphology (ameboid microglial morphology) was more evident in APP/PS1 mice treated with N-AS than APP/PS1 mice treated with vehicle, indicating increased Aβ phagocytosis (Supplementary Fig. 5b). Next because the immunohistochemistry analysis could not functionally distinguish between increased phagocytosis and impaired degradation, we analyzed the expression of Aβ degrading enzymes, including neprilysin (NEP), matrix metallopeptidase 9 (MMP9), and insulin-degrading enzyme (IDE), in WT, APP/PS1 mice injected with vehicle and APP/PS1 injected with N-AS mice. These enzymes remained unchanged between the groups, whereas CD36, which is known to be increased in phagocytic microglia[16], was enhanced in APP/PS1 mice treated with N-AS (Supplementary Fig. 5c). Finally, in Aβ plaque morphometric analysis, APP/PS1 mice treated with N-AS, rather than APP/PS1 mice treated with vehicle, showed significantly increased number of small (<25 μm) plaques, and a significant reduction of medium (25–50 μm) and large-sized (>50 μm) plaques, indicating that the outer parts of the Aβ aggregates was phagocytosed by microglia (Supplementary Fig. 5d). Overall, these results demonstrated that N-AS-triggered SPMs elevated the phagocytic capacity of microglia, and enhanced the microglial Aβ phagocytic function in APP/PS1 mice.

SPMs can stimulate the resolution of inflammation by regulating immune cells, such as limiting neutrophil infiltration and enhancing the clearance of debris by macrophages[6,7], and we also confirmed that N-AS-triggered SPMs upregulated gene expression characteristic of a positive immune response using RNA seq. To identify the effects of N-AS-triggered SPMs by microglia on the immune system, we also examined brain and blood immune cells in WT, APP/PS1 and APP/PS1 injected with N-AS mice. The APP/PS1 mice showed increased infiltration of pro-inflammatory macrophages, monocytes, neutrophils, and CD4$^+$ T cells into the brain compared with WT mice (Fig. 5c–g). Elevated levels of macrophages, pro-inflammatory monocytes, and CD4$^+$ T cells, and decreased levels of anti-inflammatory monocytes were also present in blood of APP/PS1 mice compared with WT (Supplementary Fig. 5e–i), similar to findings in a previous study[17]. These blood-derived immune cells might be promoted by Aβ, and have been shown to contribute to disease pathogenesis[17]. Interestingly, the numbers of pro-inflammatory macrophages, monocytes, and neutrophils were reduced, and the numbers of anti-inflammatory macrophages, which would facilitate clearance of cerebral Aβ, were increased in the brains of N-AS-treated APP/PS1 mice compared with vehicle-treated APP/PS1 mice (Fig. 5c–g). Further, the decreased macrophages and pro-inflammatory monocytes, and increased anti-inflammatory monocytes, were also observed in the blood of N-AS-treated APP/PS1 mice compared with vehicle-treated APP/PS1 mice, although neutrophils showed no differences between the groups (Supplementary Fig. 5e–i). These results indicated that

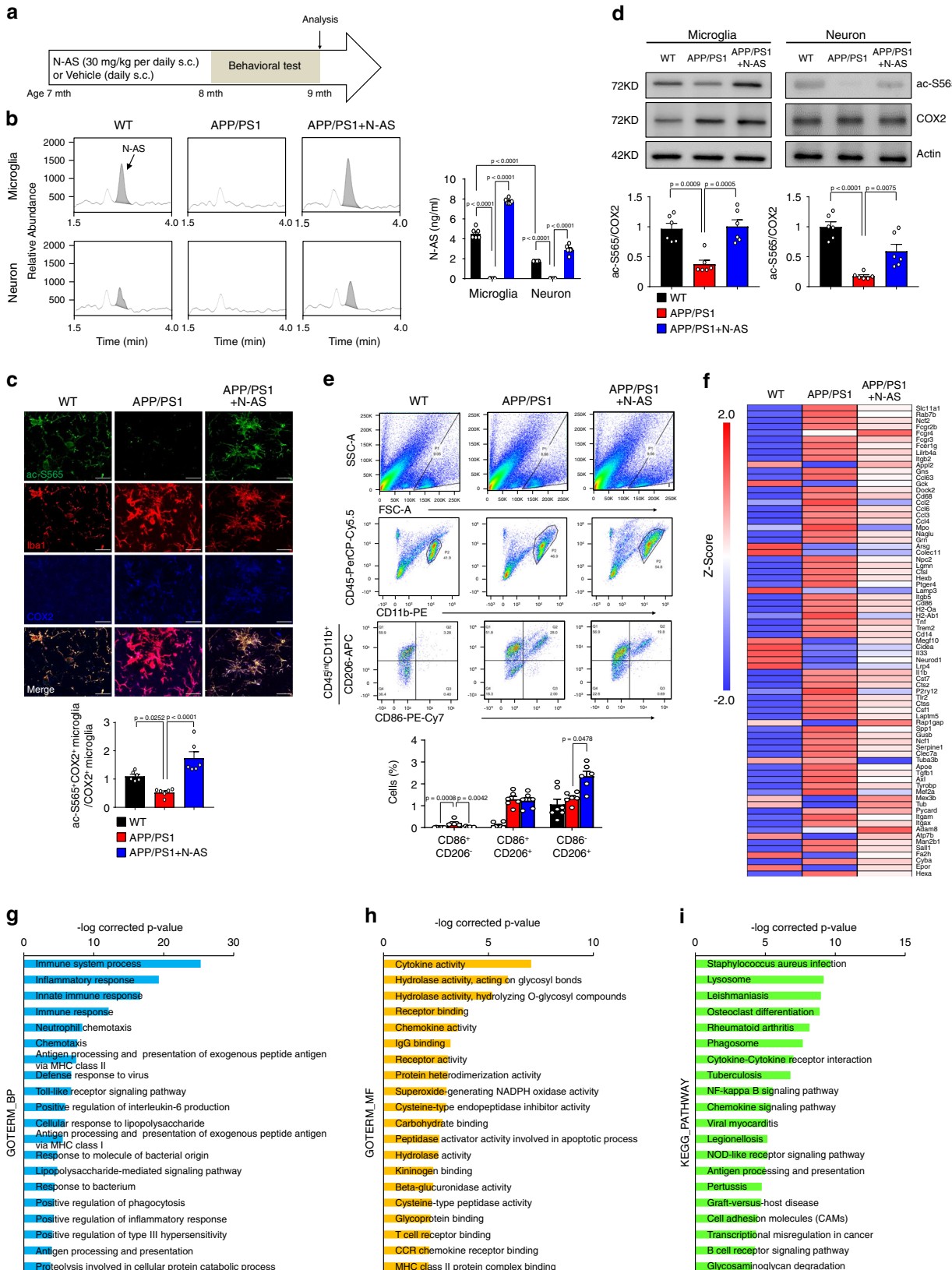

N-AS-triggered SPMs in microglia positively regulated the immune system of brain and blood, although lymphocytes, such as CD4+ T cells and B220+ B cells, did not change. Our results supported the concept that N-AS-triggered SPMs modulated microglial Aβ phagocytic function and resolved neuroinflammation via regulation of the immune system, suggesting the potential therapeutic possibilities of N-AS as a direct microglial regulator and immunomodulator in neuroinflammatory diseases, including AD.

**N-AS-triggered SPMs improve AD pathology in APP/PS1 mice.** Finally, to determine whether microglia regulation by

**Fig. 4 Decreased N-AS in APP/PS1 microglia is restored with N-AS treatment, converting neurodegenerative microglia to phagocytic microglia via N-AS-triggered SPMs. a** Scheme of experimental procedure. Daily administration of N-AS or vehicle subcutaneously to 7-months-old APP/PS1 mice for 8 weeks. **b** Representative chromatograms of N-AS and quantification in microglia and neuron derived from WT, APP/PS1 and APP/PS1 mice treated by N-AS ($n = 6$ per group). **c** Representative photomicrograph and quantification and quantification of microglia (Iba1, red) with ac-S565 (green) and COX2 (blue) in cortex of WT, APP/PS1, and APP/PS1 treated with N-AS mice. Scale bars, 20 μm ($n = 6$ per group). **d** Western blot analysis for ac-S565 and total COX2 in microglia and neuron derived from WT, APP/PS1, and APP/PS1 injected with N-AS mice ($n = 6$ per group). **e** Top, gating strategy for detection of $CD86^+CD206^-$(pro-inflammatory microglia), $CD86^+CD206^+$, and $CD86^-CD206^+$ (anti-inflammatory microglia) cells within brain cell populations. Bottom, graph displaying the calculated percentage of $CD86^+CD206^-$(pro-inflammatory microglia), $CD86^+CD206^+$, and $CD86^-CD206^+$ (anti-inflammatory microglia) cells in WT, APP/PS1, and APP/PS1 treated with N-AS mice ($n = 6$–7 mice per group). **f** Heatmap analysis of RNA seq expression data of microglia isolated WT, APP/PS1, and APP/PS1 injected with N-AS mice ($n = 3$–4 per group; each sample is a pool of two mice.). **g–i** GO term enrichment analysis of biological process (GOTERM_BP) (**g**), molecular function (GOTERM_MF) (**h**), and KEGG pathway enrichment analysis (**i**) was performed using DAVID Bioinformatics Resources 6.8. All data analysis was done at 9-months-old mice. **b–e** One-way analysis of variance. All error bars indicate s.e.m. Source data are provided as a Source data file.

N-AS-triggered SPMs in APP/PS1 AD mice affected pathology, we first determined the Aβ profile in APP/PS1 mice treated with vehicle and APP/PS1 treated with N-AS mice. Thioflavin S (ThioS) staining, immunofluorescence of 6E10, Aβ40, and Aβ42, and ELISA of Aβ40 and Aβ42 showed significantly lower Aβ levels in the APP/PS1 mice treated with N-AS compared to APP/PS1 mice treated with vehicle (Fig. 6a–e). In APP/PS1 mice treated with N-AS, cerebral amyloid angiopathy and tau hyperphosphorylation also were reduced (Fig. 6f, g). Moreover, the inflammatory response was improved in the APP/PS1 mice treated with N-AS compared to APP/PS1 mice treated with vehicle (Fig. 6h–j). Overall, these data indicated that N-AS-induced SPMs reduced Aβ plaque burden and neuroinflammation via upregulating phagocytic microglia and positive regulation of the immune response.

Next, to investigate neuropathology improvement by microglia regulation of N-AS-triggered SPMs, we performed immunolabeling for synaptophysin, MAP2, synapsin1, and PSD95 in WT, APP/PS1, and N-AS-treated APP/PS1 mice. The labeling densities of synaptophysin, MAP2, synapsin1, and PSD95 were reduced in APP/PS1 mice compared with those in WT mice, However, the expression levels of synaptophysin, MAP2, synapsin1, and PSD95 were restored in APP/PS1 mice treated with N-AS compared with those in APP/PS1 mice (Supplementary Fig. 7a–d). We also performed the Morris water maze test and fear conditioning in WT, APP/PS1 injected with vehicle and APP/PS1 injected with N-AS mice to examine changes in learning and memory. The APP/PS1 mice treated with vehicle showed severe deficits in memory formation, while APP/PS1 mice treated with N-AS were largely protected from this defect (Fig. 6k–q and Supplementary Fig. 7e). For locomotion and spontaneous activity, open field and light-dark tests were used. APP/PS1 mice treated with N-AS showed improved locomotion and spontaneous activity in the open field test compared with APP/PS1 mice, although the light–dark test did not differ between the groups (Supplementary Fig. 7f–n). These results showed that N-AS-triggered SPMs restored neuropathology and led to behavioral improvement in APP/PS1, which might be caused by increased Aβ phagocytosis via N-AS-treated microglia. However, apoptosis was not changed in the N-AS treated APP/PS1 mice compared with the vehicle treated APP/PS1 mice, although N-AS has been used extensively to induce apoptosis of tumors[18,19] (Supplementary Fig. 7o). Unlike nonsteroidal anti-inflammatory drugs including aspirin[20], the mRNA and protein levels of COX2 did not vary between APP/PS1 mice and N-AS-treated APP/PS1 mice, indicating that reduction of COX2 expression was not responsible for the restoration of AD pathology (Supplementary Fig. 7p, q). To confirm the findings above, we repeated these studies in another mouse model of AD, 5×FAD mice, and found that N-AS treatment similarly improved AD pathology of 5×FAD mice, including Aβ plaque deposit, Aβ plaque phagocytosis,

neuroinflammation, and synapse activity (Supplementary Fig. 8). Collectively, these results indicated that N-AS treatment attenuated AD pathology by regulating microglia via N-AS-triggered SPMs, but not by inducing apoptosis and decreasing COX2 expression.

## Discussion
Here, we investigated the specific mechanism of SphK1-mediated COX2 acetylation, which led to production of SPMs and resolution of neuroinflammation in microglia of AD. In SphK1-mediated COX2 acetylation, acetyl-CoA bound the ATP binding pocket of SphK1, and the acetyl group of acetyl-CoA was transferred to sphingosine, resulting in the generation of N-AS. In turn, N-AS then acetylated residue S565 of COX2, forming an intermediate between SphK1 and COX2. N-AS-acetylated COX2 led to production of SPMs, including 15R-LXA$_4$, RvE1, and RvD1, in the presence of 5-LOX, suggesting the potential pathological roles and therapeutic use of N-AS in inflammatory diseases (Supplementary Fig. 9a). Interestingly, we demonstrated that the reduction of N-AS generation is found mainly in microglia of the AD brain, leading to decreased acetyl-S565 COX2 and SPM production. The decreased N-AS-triggered SPMs caused failure of neuroinflammatory resolution and reduced phagocytosis ability, leading to an increase in immune cell trafficking and plaque burden, which exacerbated AD pathology. Moreover, N-AS administration into AD mice increased N-AS-triggered SPMs by S565 acetylation of COX2, especially in microglia, resulting in increase of phagocytic microglia. The phagocytic microglia led to resolution of neuroinflammation and an increase of Aβ phagocytosis, ameliorating AD pathology (Supplementary Fig. 9b). Collectively, our findings establish a biosynthetic mechanism and role of N-AS in AD pathogenesis, and established "proof-of-concept" for N-AS treatment in neuroinflammatory diseases including AD.

We previously demonstrated the role of neuronal SphK1-acetyltransferase activity on COX2 and showed that it was reduced in AD neurons, exacerbating AD pathology[11]. However, the mechanism of SphK1-mediated COX2 acetylation remained poorly understood, as well as its role in the pathogenesis of AD. In this study, we established N-AS as intermediate of SphK1-mediated COX2 acetylation. The N-AS was generated in neurons and microglia, which was abrogated by Aβ in the AD environment. The reduction of N-AS generation in AD neurons was caused by decreased SphK1 activity, while reduced N-AS synthesis in AD microglia was caused by deficient acetyl-CoA. Of note, a reduction of acetyl-CoA was previously found to be associated with mitochondrial dysfunction, such as inhibited pyruvate dehydrogenase complex activity, through oxidative damage by Aβ in the brains of AD 2576Tg and 3xTg mice, leading to defected microglia[21,22]. However, the correlation of microglia dysfunction

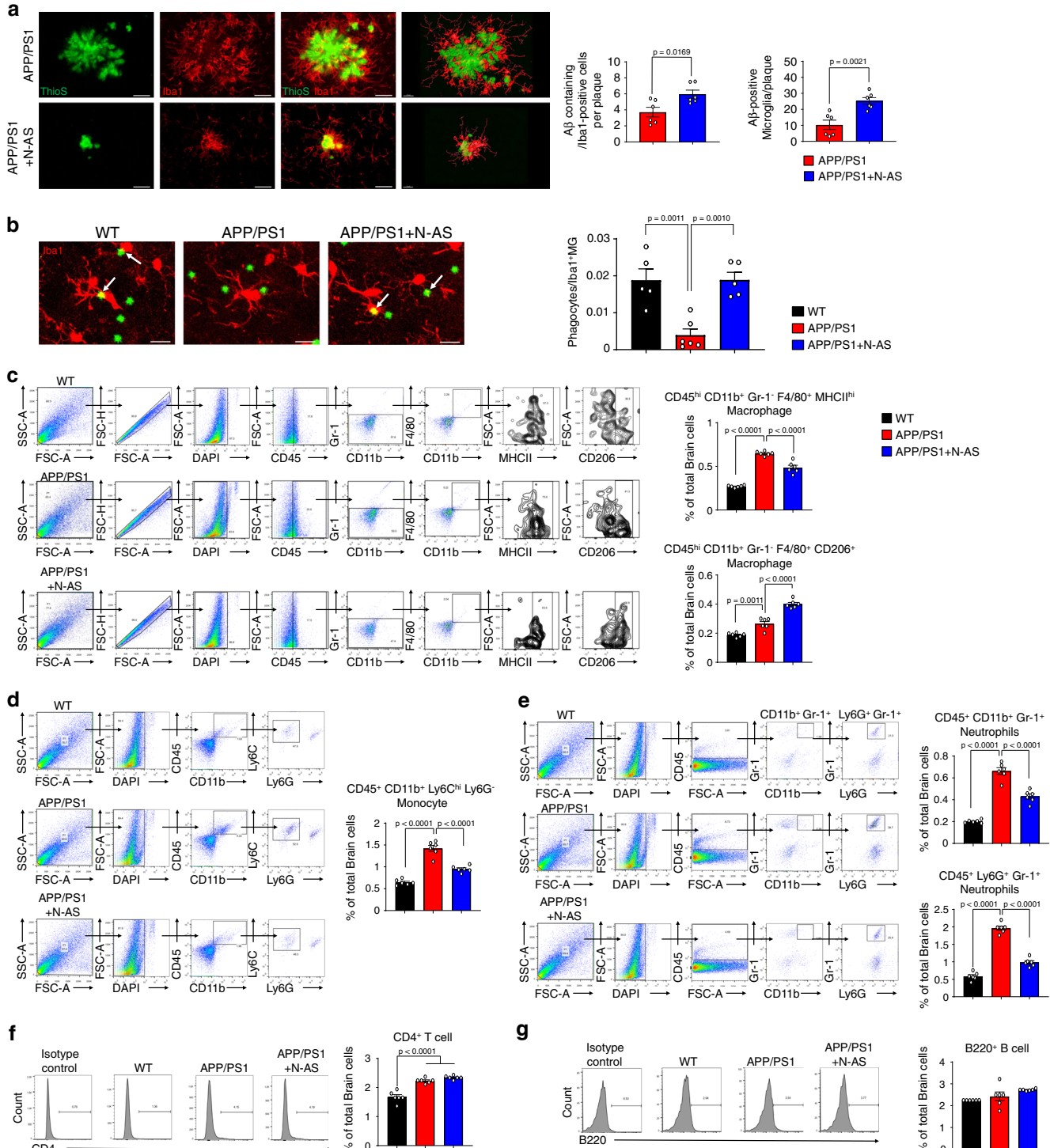

and acetyl-CoA metabolism has not been fully understood. In our study, decreased N-AS generation by deficient acetyl-CoA in AD microglia induced reduction of COX2 acetylation and SPM expression, resulting in neurodegenerative microglia. Based on previous studies and our results, we propose that deficient acetyl-CoA metabolism by Aβ destroys microglia in AD through reduction of N-AS generation.

Although there is extensive literature supporting the importance of microglial regulation[23–25] in the pathogenesis of AD, the precise regulator(s) of microglia remains to be explored in AD. The SPMs, such as RvD1 and LXA₄, play an important role in the

maintenance of phagocytic microglia[26], and are reduced in AD brain, suggesting the possibility of SPMs as microglial regulators in AD. Further, in a previous study, we reported the loss of SphK1 activity and reduced SPM secretion in AD neurons, which led to defective microglial phagocytosis and dysfunction of inflammation resolution[11], indicating the indirect regulation of microglia by neurons. In this study, our results showed that N-AS was mainly observed in microglia of N-AS-treated APP/PS1 mice rather than neurons, and directly regulated microglia via N-AS-triggered SPMs. The N-AS-triggered SPMs derived from microglia upregulated several reactive microglial genes linked to

**Fig. 5 N-AS-triggered SPMs improves phagocytic microglial function and the immune system in APP/PS1 mice brain. a** Colocalization of microglia (Iba1, red) with Aβ (ThioS, green) and quantification in cortex of APP/PS1 injected with vehicle and APP/PS1 injected with N-AS mice. Scale bars, 20 μm; 3D reconstruction from confocal image stacks scale bars, 15 μm ($n = 6$ mice per group). **b** Left, representative photomicrograph of live slice section incubated with fluorescent beads (green) in WT, APP/PS1, and APP/PS1 injected with N-AS mice. Scale bar, 10 μm. White arrow point to phagocytic microglia with fluorescent beads. Right, quantification of the number of microglial phagocytes normalized to the total number of microglia ($n = 5$–6 mice per group). **c** Left, gating strategy for detection of CD45$^{hi}$CD11b$^+$Gr-1$^-$F4/80$^+$MHCII$^{hi}$ (Pro-inflammatory macrophage) and of CD45$^{hi}$CD11b$^+$Gr-1$^-$F4/80$^+$CD206$^{hi}$ (Anti-inflammatory macrophage) cells within brain cell populations. Right, graph displaying the calculated percentage of CD45$^{hi}$CD11b$^+$Gr-1$^-$F4/80$^+$MHCII$^{hi}$ (Pro-inflammatory macrophage) and of CD45$^{hi}$CD11b$^+$Gr-1$^-$F4/80$^+$CD206$^{hi}$ (Anti-inflammatory macrophage) cells in WT, APP/PS1, and APP/PS1 treated with N-AS mice brain ($n = 6$ mice per group). **d** Left, gating strategy for detection of CD45$^+$CD11b$^+$Ly6C$^{hi}$Ly6G$^-$ monocyte within brain cell populations. Right, graph displaying the calculated percentage of CD45$^+$CD11b$^+$Ly6C$^{hi}$Ly6G$^-$ monocyte in WT, APP/PS1, and APP/PS1 treated with N-AS mice brain ($n = 6$ mice per group). **e** Left, gating strategy for detection of CD45$^+$CD11b$^+$Gr-1$^+$ neutrophil and CD45$^+$Ly6G$^+$Gr-1$^+$ neutrophil within brain cell populations. Right, graph displaying the calculated percentage of CD45$^+$CD11b$^+$Gr-1$^+$ neutrophil and CD45$^+$Ly6G$^+$Gr-1$^+$ neutrophil in WT, APP/PS1, and APP/PS1 treated with N-AS mice brain ($n = 6$ mice per group). **f** Left, histograms are representative of CD4$^+$ T cell proliferation at brain. Right, graph displaying the calculated percentage of CD4$^+$ T cell in WT, APP/PS1, and APP/PS1 treated with N-AS mice brain ($n = 6$ mice per group). **g** Left, histograms are representative of B220$^+$ B proliferation at brain. Right, graph displaying the calculated percentage of B220$^+$ B cell in WT, APP/PS1, and APP/PS1 treated with N-AS mice brain ($n = 6$ mice per group).All data analysis was performed on 9-months-old mice. **a–g** One-way analysis of variance, Tukey's post hoc test. All error bars indicate s.e.m. Source data are provided as a Source data file.

phagocytosis, and downregulated several inflammatory and immune molecules in microglia of N-AS-treated APP/PS1 mice. We also confirmed that N-AS-triggered SPMs derived from microglia resolves neuroinflammation via regulation of the immune system, similar to the function of SPMs reported in previous studies[6,7]. Furthermore, we confirmed that N-AS-triggered SPMs produced from microglia directly mediate the effects seen after administration of N-AS. When 5-LOX inhibitor, zileuton, was incubated with N-AS in Aβ-treated human microglia, the production of N-AS-triggered SPMs and restoration of phagocytic capacity was inhibited, indicating that N-AS-triggered SPMs directly mediate the positive effects in microglia. However, because the zileuton also reduce formation of leukotriene (e.g., LTB$_4$) and prostaglandin (e.g., PGE$_2$) production[27], we cannot completely rule out the possibility that blocking 5-LOX may result in the reduction of other lipid mediators that may have effects on phagocytosis. Our results showed that zileuton treatment reduced PGE$_2$, as well as SPMs, although LTB$_4$ was not detected. Both SPMs and PGE$_2$, which decreased by zileuton, play an important role in phagocytic ability of microglia, and phagocytosis is restored through the increase of SPMs production and inhibition of PGE$_2$ synthesis[11,24]. In our study, although the levels of PGE$_2$ were decreased, the phagocytic ability was not restored by N-AS in Aβ-treated microglia with zileuton. These results suggest the N-AS-triggered SPMs, rather than the other lipid mediators, were more responsible direct regulation of microglia including phagocytosis in AD environment. Notably, N-AS-triggered SPMs, as well as N-AS generation and acetyl-S565 COX2, in microglia were higher than those in neurons, suggesting that the N-AS-triggered SPMs derived from microglia might play an important role rather than those derived from neurons. Collectively, these results indicated that direct microglial regulation via N-AS-triggered SPMs derived from microglia might be more responsible for AD pathogenesis than indirect regulation of microglia thorough neurons, suggesting N-AS-triggered SPMs derived from microglia as direct regulators of pathology in AD.

In AD diagnosis, positron emission tomography (PET) imaging procedures are used for detecting Aβ and tau material when molecular aggregates are deposited in the brain at high concentration in the late stages of AD pathology. However, it is very difficult to diagnosis the disease before the protein aggregates[28]. The neuroinflammation in the AD pathology starts at very early stages, most likely before amyloid plaques are formed, suggesting it may be a promising therapeutic target and useful in the diagnosis of AD at preclinical stages[28]. In fact, use of radiolabeled translocator protein (TSPO, first described as the peripheral benzodiazepine receptor) ligands to quantify the expression of the microglial TSPO receptor has been applied for the diagnosis of neuroinflammation[28,29]. Similar to radiolabeled TSPO ligands, N-AS might be considered to diagnose neuroinflammation in early AD stages, in addition to its potential therapeutic use, based on two results: (1) In PK profiling of N-AS, a high concentration of N-AS was maintained in plasma and brain, and N-AS had high-distribution to the brain; and (2) N-AS had high binding affinity and induced high-level acetylation of the elevated COX2, prior to observing Aβ deposits in APP/PS1 microglia. Despite the potential of N-AS as a therapeutic, it is also important to recognize that this therapeutic use of N-AS may be limited by its poor solubility and low BA. There is also the potential of N-AS to induce apoptosis, which has been found in other studies[18,19]. Therefore, additional studies are warranted to investigate derivatives of N-AS that might be potentially useful for neuroinflammatory diseases including AD.

## Methods

**Molecular docking**. Molecular modeling studies were performed using Discovery Studio Programs (2018, Accelrys) and figures were generated using PyMOL programs. The 3D coordinates of human SphK1 complexed with ADP were obtained from the protein data bank for docking studies (PDB entry 3VZD). 3VZD represents a tetrameric crystal structure of SphK1 that includes six monomers (A-F). Monomer C was chosen for the simulations based on its displacement property. ADP, which binds between the N-terminal and C-terminal domains, was removed from the complex. Preparation of the protein was performed with the Prepare Protein protocol and a binding site was created using the Define and Edit Binding Site protocol (default parameters were used). Energy minimization of ligands was performed by the Clean Geometry module in Discovery Studio, and conformations were generated using the Generate Conformations protocol and the Best method (other parameters were kept at the default values). Top ten conformations with the highest degree of similarity to ADP were chosen for docking simulation. Docking of the ligands (acetyl-CoA, sphingosine, and ATP) was performed using the CDOCKER protocol, and analysis of ligand interactions at the binding site was performed using the ligand Interactions tool followed by visual inspection.

**S1P assays**. We confirmed S1P formation through SphK1 enzymatic activity measurements as previously described[11] using a UPLC (Ultra performance liquid chromatography) system (Waters). Briefly, 3 μl of SphK1 (Cayman, 10348) were mixed with 3 μl of assay buffer (200 μM NBD-sphingosine (Avanti Polar Lipids), 100 mM of HEPES buffer, pH 7.2, 10 mM MgCl$_2$, 200 mM Semicarbazide, 1 mM 4-deoxypyridoxin, 2 mM Dithiothreitol, and 0.2% Igepal CA-630, all from Sigma-Aldrich) in the presence of 0–50 μM acetyl-CoA (Sigma-Aldrich, A2056), and the mixture were preincubated at 37 °C for 10 min. After pre-incubation, 1 mM ATP (Amresco, 0220) was added the mixture, and it was incubated at 37 °C for 1 h. The hydrolysis reactions were stopped by adding 53 μl of ethanol, and centrifuged at 15,493×g for 5 min. Thirty microliter of the supernatant was then transferred to a sampling glass vial and 5 μl was applied onto a UPLC system for analysis. S1P formation was followed as phosphorylation of (7-nitro-2-1,3-benzoxadiazol-4-yl)-d-erythro (NBD)-sphingosine

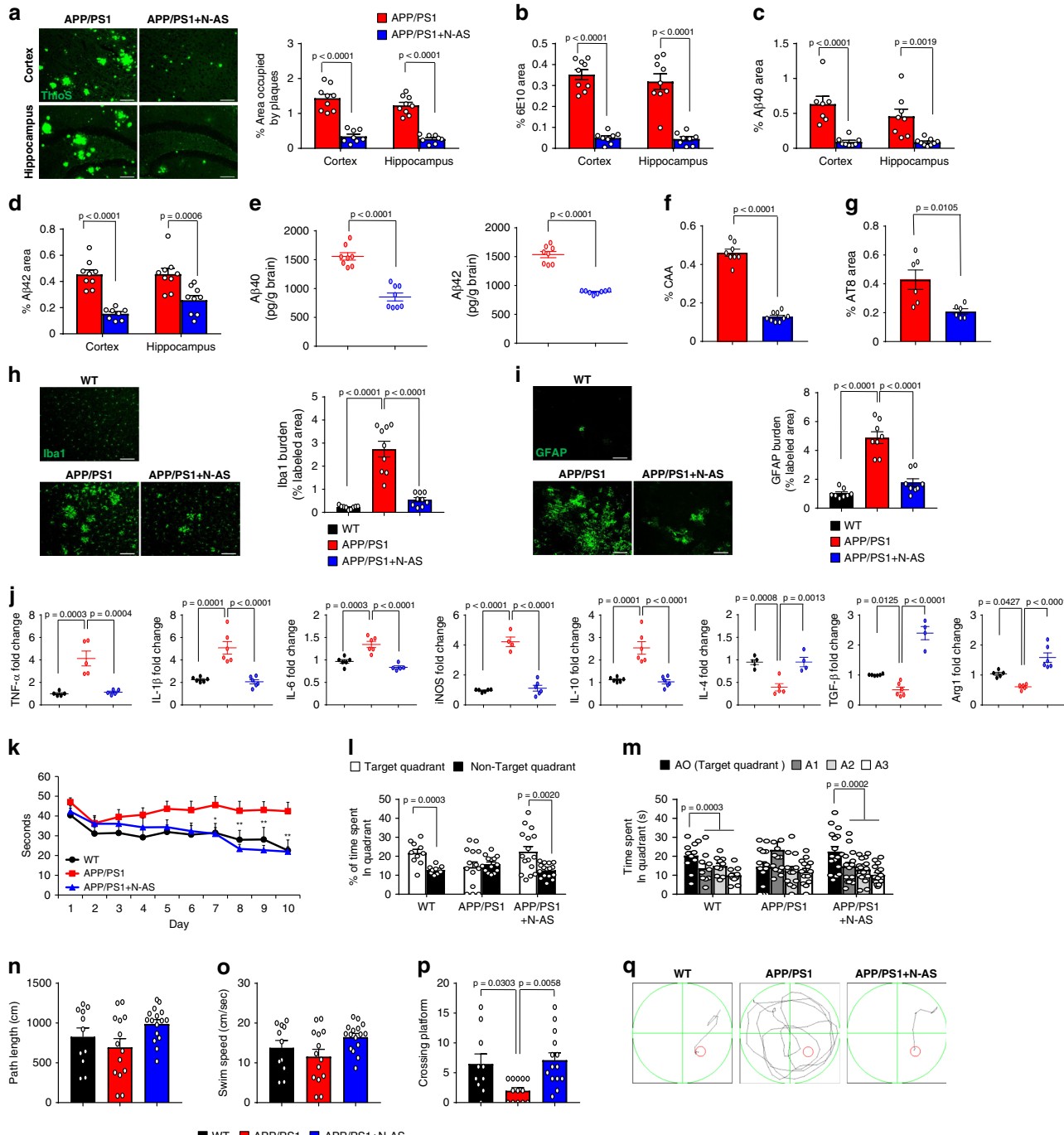

**Fig. 6 Administration of N-AS reduces amyloid pathology and restores cognitive function in APP/PS1 mice by N-AS-triggered SPMs. a** Left, representative immunofluorescence images of thioflavin S (ThoS, Aβ plaques) in cortex and hippocampus of APP/PS1 and APP/PS1 treated with N-AS mice. Scale bars, 100 μm. Right quantification of area occupied by Aβ plaques (APP/PS1 mice, n = 9; APP/PS1 + N-AS mice, n = 8). **b** Quantification of 6E10 in cortex and hippocampus of APP/PS1 and APP/PS1 treated with N-AS mice (n = 8–9 mice per group). **c–e** Analysis of Aβ40 and Aβ42 depositions from the mice brain samples of the APP/PS1 and APP/PS1 treated with N-AS mice using immunofluorescence staining (**c, d** n = 7–9 mice per group) and ELISA kits (**e** n = 8 mice per group). **f, g** Quantification of amyloid angiopathy (**f** n = 8–9 mice per group, CAA) and tau hyperphosphorylation (**g** n = 6 mice per group, AT8) in cortex of APP/PS1 and APP/PS1 treated with N-AS mice. **h** Left, immunofluorescence images of microglia (Iba1) in cortex of WT, APP/PS1, and APP/PS1 treated with N-AS mice brain. Scale bars, 100 μm. Right, quantification of Iba1+ microglia in WT, APP/PS1, and APP/PS1 treated with N-AS (n = 9 mice per group). **i** Left, immunofluorescence images of astrocyte (GFAP) in cortex of WT, APP/PS1, and APP/PS1 treated with N-AS mice brain. Scale bars, 100 μm. Right, quantification of GFAP+ astrocyte in WT, APP/PS1, and APP/PS1 treated with N-AS (n = 8 mice per group). **j** mRNA levels of inflammatory markers in cortex of WT, APP/PS1, and APP/PS1 treated with N-AS mice brain (n = 4–6 per group). Pro-inflammatory marker: TNF-α, IL-1β, IL-6, and iNOS, Immunoregulatory cytokine: IL-10, Anti-inflammatory marker: IL-4, TGF-β, and Arg1. **k** Learning and memory was assessed by Morris water maze test in the WT (n = 10), APP/PS1 (n = 14), and APP/PS1 injected with N-AS (n = 16) mice. **l–q** At probe trial day 11 of Morris water maze test, time spent in target platform (**l**) and other quadrants (**m**) was measured. Path length (**n**) and swim speed (**o**) analyzed. **p** The number of times each animal entered the small target zone during the 60 s probe trial. **q** Representative swimming paths at day 10 of training. **a–g, l** Student's t-test. **h–k, m–p** One-way analysis of variance, Tukey's post hoc test. All error bars indicate s.e.m. Source data are provided as a Source data file.

(Avanti Polar Lipids) to NBD-S1P. Quantification was achieved by comparison with NBD-S1P (Avanti Polar Lipids) standards.

**Mutagenesis of the SphK1 and COX2 proteins.** Single point mutations in SphK1 (R56A, R57A, R24A, R185A, D178A, and F192A) and COX2 (S565A, N181A, T564A, and S567A) were generated using the In-Fusion Cloning Kits (Clontech) and ORF cDNA expression plasmids of human SphK1 and COX2 gene (Sino Biological Inc, HG15679-NH and HG12036-NH), according to the manufacturer's instructions. The forward and reverse primers described in Supplementary Table 4. The SphK1 WT, SphK1 mutants, COX2 WT, and COX2 mutant plasmids were transformed in the One shot TOP 10 competent cell (Invitrogen). The bacteria were grown in Luria-Bertani (LB, Invitrogen) plates supplemented with 50 μg ml⁻¹ kanamycin (Sigma-Aldrich) at 37 °C. The bacterial cell pellets were resuspended in lysis buffer containing 20 mM sodium phosphate (GE Healthcare), 500 mM NaCl (GE Healthcare), 20 mM imidazole (pH 7.4) (GE Healthcare), 0.2 mg ml⁻¹ lysozyme (Sigma-Aldrich), 20 μg ml⁻¹ DNase I (Roche), 1 mM MgCl₂ (Sigma-Aldrich), and 1 mM PMSF (Sigma-Aldrich) before being lysed by sonication and clarified by centrifugation at 15,493×g for 10 min. Proteins were purified from the soluble fraction using His-Spin Trap columns (GE Healthcare) and the binding and acetylation assays were performed.

**Enzyme kinetics.** The acetyl-CoA binding activity of SphK1 or the SphK1 WT and SphK1 mutants (R56A, R57A, R24A, and R185A) in the presence of ATP was analyzed by filter binding assays[11]. Briefly, SphK1 WT and mutant enzymes (1 mU) were resuspended in 100 μl reaction buffer (20 mM HEPES; pH 7.4, 50 mM NaCl, 10 mM MgCl₂, 1 mM EGTA, 0.02% Triton-X100, and 100 μM sphingosine, all from Sigma-Aldrich). Reactions were initiated by adding [³H] acetyl-CoA (1–200 μM (0.1–20 μCi), Perkin-Elmer, NET290050UC) and incubated at 37 °C for 1 h. The reactions were terminated by adding 100 μl of ice-cold reaction buffer containing 5 mM cold acetyl-CoA and immediately filtered through P30 filtermat (Perkin-Elmer), followed by five washes. The N-AS binding activity of COX2 WT and COX2 mutants (S565A, N181A, T564A, and S567A) was analyzed similarly to the acetyl-CoA binding activity of SphK1. Briefly, COX2 WT and mutant enzymes (1 mU) were resuspended in 100 μl reaction buffer (20 mM HEPES; pH 7.4, 50 mM NaCl, 10 mM MgCl₂, 1 mM EGTA, and 0.02% Triton-X100, all from Sigma-Aldrich). Reactions were initiated by adding [¹⁴C] N-AS (1–200 μM (0.1–20 μCi), ARC UK Ltd, ARC-1024) and incubated at 37 °C for 1 h. The reactions were terminated by adding 100 μl of ice-cold reaction buffer containing 5 mM cold N-AS and immediately filtered through P30 filtermat (Perkin-Elmer), followed by five washes.

Incorporation of SphK1/acetyl-CoA and COX2/N-AS was analyzed using a Micro Beta 2 liquid scintillation counter, respectively. The binding velocity was plotted and the $K_{cat}$ and $K_M$ values were calculated from the Michaelis-Menten analysis using the Sigma-plot program (ver 10.0, Systat software Inc.).

**In vitro and in vivo acetylation assays.** Acetylation assays were performed according to previously described procedures with minor modifications[11]. To confirm the SphK1-mediated COX2 acetylation, 10 μg COX2 (LSBio, LS-G21094 and Cayman, 60122) was incubated with [¹⁴C] acetyl-CoA (2 μCi, Perkin-Elmer, NEC313050UC) and SphK1 (WT or mutants) in the presence of sphingosine (100 μM, Sigma-Aldrich, S7049) and/or ATP (0–1000 μM, Amresco, 0220) at 37 °C for 2 h. To confirm the N-AS-mediated COX2 acetylation, 10 μg COX2 (WT or mutants) was used incubated with [¹⁴C] N-AS (2 μCi, ARC UK Ltd, ARC-1024) at 37 °C for 2 h. [¹⁴C] aspirin (2 μCi, ARC UK Ltd, ARC-0473) was conducted as the positive control. The reactions were terminated by placing the reaction tubes in ice for 5 min. To evaluate in vivo neuronal and microglial COX2 acetylation, 10 μCi [¹⁴C] N-AS was orally administrated to 9-months-old WT and APP/PS1 mice. After 1 h, neuron and microglia were isolated and sonicated in culture media. To immunoprecipitate acetylated COX2, the sample was incubated with COX2 antibody (Abcam, ab15191) at 4 °C for 24 h and precipitated by adding 50 μl of protein-A sepharose (GE Healthcare). Acetylated COX2 proteins bound to the beads were analyzed using a Tri-Carb 3110TR liquid scintillation counter (Perkin-Elmer).

**General chemical synthesis.** All the reactions were carried out in oven dried glassware under nitrogen atmosphere with freshly distilled dry solvents under anhydrous conditions unless otherwise indicated. Flash column chromatography was performed with Silica Flash P60 silica gel (230–400 mesh). All reagents were obtained from commercial sources and were used without further purification. Proton (¹H) and carbon (¹³C) NMR spectra were recorded on a 400/100 MHz Agilent 400 M FT-NMR spectrometer or 400/100 MHz Bruker Advance III HD FT-NMR spectrometer. NMR solvents were obtained from Cambridge Isotope Laboratories and the residual solvent signals were taken as the reference (CDCl₃, 7.26 ppm or DMSO-d6, 2.50 ppm for ¹H NMR spectra and CDCl₃, 77.0 ppm or DMSO-d6, 39.52 ppm for ¹³C NMR spectra). For annotated NMR spectra see Data S1. Mass analysis were carried out using Advion Expression CMS mass spectrometer and LC-ELSD analysis was carried out using Agilent 1260 Infinity ELSD coupled with Agilent 1220 HPLC. High resolution mass analysis was performed with JOEL AccuTOF 4G+DART-HRMS.

**Synthesis of acetyl sphingosines.** (S)−3-Hydroxy-1-methoxy-1-oxopropan-2-aminium chloride (**1a**)

To a solution of L-Serine (530 mg, 5 mmol) in methanol (5 ml) at 0 °C was added SOCl₂ (1.27 ml, 17.5 mmol) dropwise. The mixture was stirred for overnight at 60 °C and concentrated in vacuo. The crude material was washed with ethanol (2 ml) to yield **1a** (775 mg, 4.98 mmol) as a white solid.
¹H NMR (400 MHz, DMSO-d₆): δ 8.53 (br s, 3H), 5.61 (t, J = 5.1 Hz, 1H), 4.11 (t, J = 3.6 Hz, 1H), 3.87–3.77 (m, 2H), 3.74 (s, 3H).
¹³C NMR (100 MHz, DMSO-d₆): δ 168.47, 59.45, 54.42, 52.75.
MS (APCI, M+H⁺): m/z found 120.0.
Methyl (tert-butoxycarbonyl)-L-serinate (**1b**)

To a solution of **1a** (770 mg, 4.95 mmol) in DCM (16 ml) at 0 °C was added triethylamine (1.73 ml, 12.38 mmol) and Boc₂O (1.42 g, 6.44 mmol). The mixture was slowly warmed to room temperature (RT), and stirred overnight. The mixture was concentrated in vacuo and purified by flash chromatography (10–30% ethyl acetate in hexanes) to yield **1b** (997 mg, 4.55 mmol) as a yellow oil.
¹H NMR (400 MHz, CDCl₃): δ 5.42 (br s, 1H), 4.40 (br s, 1H), 4.00–3.88 (m, 2H), 3.79 (s, 3 H), 2.19 (br s, 1H), 1.46 (s, 9H).
¹³C NMR (100 MHz, CDCl₃): δ 171.33, 155.72, 80.29, 63.48, 55.69, 52.59, 28.26.
MS (APCI, M+H⁺): m/z found 220.0.
3-(tert-Butyl) 4-methyl (S)-2,2-dimethyloxazolidine-3,4-dicarboxylate (**1c**)

To a solution of **1b** (965 mg, 4.40 mmol) in toluene (11 ml) at RT was added p-toluenesulfonic acid monohydrate (17 mg, 0.088 mmol), and 2,2-dimethoxypropane (1.1 ml, 8.85 mmol). The mixture was stirred for 2 h at 80 °C, and cooled to RT. The mixture was concentrated in vacuo, and purified by flash chromatography (5–15% ethyl acetate in hexanes) to yield **1c** (692 mg, 2.67 mmol) as clear oil.
¹H NMR (400 MHz, CDCl₃): (as rotamers) δ 4.49 and 4.38 (2dd, J = 6.8, 2.8 Hz, and J = 7.0, 3.1 Hz, 1H), 4.17–4.11 (m, 1H), 4.07–4.01 (m, 1H), 3.76 (s, 3H), 1.68–1.41 (m, 15H).
¹³C NMR (100 MHz, CDCl₃): (as rotamers) δ 171.66, 171.25, 152.07, 151.17, 95.04, 94.39, 80.86, 80.29, 66.24, 65.99, 59.26, 55.19, 52.36, 52.24, 28.33, 28.26, 26.00, 25.16, 24.93, 24.36.
MS (APCI, M+H⁺): m/z found 260.1.
tert-Butyl (R)-4-(hydroxymethyl)-2,2-dimethyloxazolidine-3-carboxylate (**1d**)

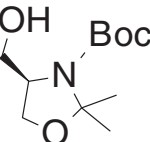

To a solution of lithium aluminum hydride (221 mg, 5.82 mmol) in THF (5 ml) at −78 °C was added a solution of **1c** (500 mg, 1.92 mmol) in THF (5 ml) dropwise. The mixture was stirred for 10 min at −78 °C and warmed to RT, stirred for 30 min. The reaction was quenched by adding 1 N NaOH (10 ml) at −78 °C. The mixture was filtered through celite, and extracted with ethyl acetate. The organic layer was dried over Na₂SO₄, filtered and concentrated. The crude material was purified by flash chromatography (20–50% ethyl acetate in hexanes) to yield **1d** (407 mg, 1.76 mmol) as a clear oil.

$^1$H NMR (400 MHz, CDCl$_3$): δ 4.13–3.98 (m, 2H), 3.79–3.69 (m, 2H), 3.64 (br s, 1H), 1.61–1.45 (m, 15H).

$^{13}$C NMR (100 MHz, CDCl$_3$): δ 154.20, 94.11, 81.16, 65.45, 65.26, 59.52, 28.37, 27.10, 24.59.

MS (APCI, M+H$^+$): *m/z* found 232.2.

tert-Butyl (S)-4-formyl-2,2-dimethyloxazolidine-3-carboxylate (**1e**)

To a solution of DMSO (355 µl, 4.93 mmol) in DCM (0.5 ml) at −78 °C was added a solution of oxalyl chloride (215 µl, 2.47 mmol) in DCM (3.5 ml) dropwise, and the mixture was stirred for 20 min. The mixture was warmed to −60 °C, and was added a solution of **1d** (380 mg, 1.64 mmol) in DCM (4 ml) dropwise over 15 min. The mixture was stirred for 20 min, and was added DIPEA (1.75 ml, 10 mmol) dropwise. The mixture was slowly warmed to 0 °C over 2 h, and was added ice-cold 1 N HCl (16 ml), and extracted with DCM. The organic layer was washed with ice-cold brine (10 ml), and was dried over Na$_2$SO$_4$, filtered and concentrated in vacuo to yield **1e** (370 mg, 1.61 mmol) as a yellow oil. The residue was used in the next step without further purification.

tert-Butyl((S)-4-((R)-1-hydroxyhexadec-2-yn-1-yl)-2,2-dimethyloxazolidine-3-carboxylate (**1f**)

To a solution of 1-pentadecyne (0.29 ml, 1.1 mmol) in THF (6 ml) at −40 °C was added n-butyllithium (0.5 ml, 1 mmol, 2 M in hexanes) dropwise, and the mixture was stirred for 30 min. To the mixture was added a solution of **1e** (115 mg, 0.50 mmol) in THF (3.5 ml) dropwise over 10 min, and the mixture was stirred for 6 h. To the mixture was added water (3 ml) dropwise and warmed to r.t. and was added brine (5 ml) and extracted with ethyl acetate. The combined organic layers were dried over Na$_2$SO$_4$ and filtered, and concentrated in vacuo. The crude material was purified by flash chromatography (5–20% ethyl acetate in hexanes) to yield **1f** (175 mg, 0.40 mmol) as a clear oil.

$^1$H NMR (400 MHz, CDCl$_3$): δ 4.74–4.68 (m, 1H), 4.54–4.48 (m, 1H), 4.16–4.04 (m, 2H), 3.91 (br s, 1H), 2.19 (td, *J* = 7.1, 1.8 Hz, 2H), 1.52–1.24 (m, 37H), 0.88 (t, *J* = 7.0, 3H).

$^{13}$C NMR (100 MHz, CDCl$_3$): δ 154.09, 94.90, 86.60, 81.19, 77.83, 65.09, 64.15, 62.83, 31.89, 29.65, 29.63, 29.62, 29.61, 29.49, 29.32, 29.11, 28.87, 28.55, 28.37, 25,74, 25.40, 22.66, 18.73, 14.08.

**MS** (APCI, M+H$^+$): *m/z* found 438.4.

tert-Butyl ((2S,3R)-1,3-dihydroxyoctadec-4-yn-2-yl)carbamate (**1g**)

To a solution of **1f** (1.36 g, 3.11 mmol) in methanol (31 ml) at 0 °C was added p-toluenesulfonic acid monohydrate (30 mg, 0.155 mmol), and the mixture was stirred for 5 min. The mixture was warmed to RT, and was stirred for 19 h. The mixture was concentrated in vacuo, and the crude material was purified by flash chromatography (20–50% ethyl acetate in hexanes) to yield **1g** (952 mg, 2.39 mmol) as a white solid.

$^1$H NMR (400 MHz, CDCl$_3$): δ 5.31 (br s, 1H), 4.60 (br s, 1H), 4.15–4.09 (m, 1H), 3.80–3.73 (m, 2H), 2.77 (br s, 1H), 2.29 (br s, 1H), 2.22 (td, *J* = 7.1, 2.0 Hz, 2H), 1.46 (s, 9H), 1.37–1.24 (m, 22H) 0.88 (t, *J* = 7.0, 3H).

$^{13}$C NMR (100 MHz, CDCl$_3$): δ 156.21, 88.13, 80.02, 77.83, 64.66, 62.82, 55.69, 31.89, 29.66, 29.63, 29.62, 29.50, 29.32, 29.10, 28.89, 28.51, 28.33, 22.66, 18.68, 14.09.

MS (APCI, M+H$^+$): *m/z* found 398.3.

tert-Butyl ((2S,3R,E)-1,3-dihydroxyoctadec-4-en-2-yl)carbamate (**1h**)

To a solution of **1g** (400 mg, 1 mmol) in THF (10 ml) at 0 °C was added sodium bis(2-methoxyethoxy)aluminum hydride (0.74 ml, 2.35 mmol, 60 wt% in toluene) dropwise. The mixture was stirred for 2 h, and slowly warmed to RT, and was stirred for 6 h. The mixture was cooled to 0 °C, and was added 1 N HCl (0.3 ml) dropwise. The mixture was filtered through celite, and concentrated in vacuo. The crude material was purified by flash chromatography (5–30% ethyl acetate in hexanes) to yield **1h** (312 mg, 0.78 mmol) as a white solid.

$^1$H NMR (400 MHz, CDCl$_3$): δ 5.82–5.75 (m, 1H), 5.53 (ddt, *J* = 15.4, 6.5, 1.4 Hz, 1H), 5.28 (br s, 1H), 4.35–4.30 (m, 1H), 3.94 (dt, *J* = 11.3, 3.7 Hz, 1H), 3.74–3.69 (m, 1H), 3.60 (br s, 1H), 2.44 (br s, 2H), 2.08–2.03 (m, 2H), 1.46 (s, 9H), 1.40–1.23 (m, 22H), 0.88 (t, *J* = 7.0, 3H).

$^{13}$C NMR (100 MHz, CDCl$_3$): δ 156.21, 134.12, 128.90, 79.78, 74.76, 62.63, 55.38, 32.28, 31.91, 29.67, 29.66, 29.64, 29.60, 29.47, 29.34, 29.20, 29.09, 28.36, 22.67, 14.10.

MS (APCI, M+H$^+$): *m/z* found 400.5.

(2S,3R,E)-2-((tert-Butoxycarbonyl)amino)-3-hydroxyoctadec-4-en-1-yl acetate (**2a**)

To a solution of **1h** (40 mg, 0.1 mmol), and 4-dimethylaminopyridine (12 mg, 0.1 mmol) in DCM (1.4 ml) at 0 °C was added a solution of acetic anhydride (10.4 µl, 0.11 mmol) in DCM (0.6 ml) dropwise. The mixture was stirred for 2 h, and was concentrated in vacuo. The crude material was purified by flash chromatography (5–15% ethyl acetate in hexanes) to yield **2a** (31 mg, 0.07 mmol) as a white solid.

$^1$H NMR (400 MHz, CDCl$_3$): δ 5.79–5.71 (m, 1H), 5.49 (dd, *J* = 15.4, 6.8 Hz, 1H), 4.85 (br s, 1H), 4.27 (dd, *J* = 11.5, 6.2 Hz), 4.19–4.11 (m, 2H), 3.90 (br s, 1H), 2.35 (br s, 1H), 2.07 (s, 3H), 2.06–2.01 (m, 2H), 1.45 (s, 9H), 1.39–1.23 (m, 22H), 0.88 (t, *J* = 7.0, 3H).

$^{13}$C NMR (100 MHz, CDCl$_3$): δ 171.04, 155.84, 134.61, 128.26, 79.77, 73.18, 63.24, 54.01, 32.28, 31.90, 29.65, 29.63, 29.57, 29.46, 29.33, 29.20, 29.06, 28.31, 22.66, 20.86, 14.09.

MS (APCI, M+H$^+$): *m/z* found 442.3.

(2S,3R,E)-1-Acetoxy-3-hydroxyoctadec-4-en-2-aminium trifluoroacetate (1-O-AS)

To a solution of **2a** (14 mg, 0.031 mmol) in DCM (0.45 ml) at 0 °C was added trifluoroacetic acid (190 µl, 2.48 mmol), and triethylsilane (15 µl, 0.093 mmol). The mixture was stirred for 2 h, and concentrated in vacuo to yield **1-O-AS** (14 mg, 0.031 mmol) as pale-yellow oil.

$^1$H NMR (400 MHz, CDCl$_3$): δ 5.91–5.83 (m, 1H), 5.44 (dd, *J* = 15.4, 6.2 Hz, 1H), 4.53 (br s, 1H), 4.34–4.26 (m, 2H), 3.54–3.49 (m, 1H), 2.09 (s, 3H), 2.07–2.02 (m, 2H), 1.41–1.22 (m, 22H), 0.88 (t, *J* = 7.0, 3H).

$^{13}$C NMR (100 MHz, CDCl$_3$): δ 171.65, 137.13, 124.65, 70.02, 60.69, 55.34, 32.20, 31.91, 29.69, 29.67, 29.65, 29.58, 29.44, 29.35, 29.24, 28.77, 22.67, 20.35, 14.09.

HRMS (DART, M+H$^+$): *m/z* calcd. for C$_{20}$H$_{40}$NO$_3$$^+$ 342.3003, found 342.3003.

tert-Butyl ((2S,3R,E)-1-((tert-butyldimethylsilyl)oxy)-3-hydroxyoctadec-4-en-2-yl)carbamate (**3a**)

To a solution of **1h** (37 mg, 0.092 mmol), and imidazole (12.5 mg, 0.184 mmol) in DCM (0.6 ml) at 0 °C was added a solution of tert-butyldimethylsilyl chloride (15.4 mg, 0.1 mmol) in DCM (0.3 ml) dropwise. The mixture was stirred for 1 h, and warmed to RT, and stirred for 0.5 h. The mixture was concentrated in vacuo, and the crude material was purified by flash chromatography (5–15% ethyl acetate in hexanes) to yield **3a** (39 mg, 0.076 mmol) as a white solid.

$^1$H NMR (400 MHz, CDCl$_3$): δ 5.80–5.72 (m, 1H), 5.50 (dd, $J = 15.4$, 5.9 Hz, 1H), 5.22 (br d, $J = 7.8$ Hz, 1H), 4.22–4.16 (m, 1H), 3.94 (dd, $J = 10.3$, 3.0 Hz, 1H), 3.78–3.71 (m, 1H), 3.57 (br s, 1H), 3.30 (br d, $J = 8.1$ Hz, 1H), 2.09–2.02 (m, 2H), 1.45 (s, 9H), 1.40–1.24 (m, 22H), 0.91–0.86 (m, 12H), 0.07 (d, $J = 1.1$ Hz, 6H).

$^{13}$C NMR (100 MHz, CDCl$_3$): δ 155.74, 133.06, 129.42, 79.39, 74.62, 63.45, 54.49, 32.30, 31.91, 29.68, 29.64, 29.62, 29.50, 29.35, 29.19, 28.38, 25.81, 22.68, 18.12, 14.10, −5.62, −5.65.

MS (APCI, M+H$^+$): $m/z$ found 514.4.

(2S,3R,E)-2-((tert-Butoxycarbonyl)amino)-1-((tert-butyldimethylsilyl)oxy) octadec-4-en-3-yl acetate (**3b**)

To a solution of **3a** (38 mg, 0.074 mmol), and 4-dimethylaminopyridine (18 mg, 0.15 mmol) in DCM (0.7 ml) at RT, was added acetic anhydride (10.5 µl, 0.11 mmol). The mixture was stirred for 1 h, and concentrated in vacuo. The crude material was purified by flash chromatography (5–10% ethyl acetate in hexanes) to yield **3b** (38 mg, 0.068 mmol) as a clear oil.

$^1$H NMR (400 MHz, CDCl$_3$): δ 5.81–5.73 (m, 1H), 5.41 (dd, $J = 15.4$, 7.6, 1H), 5.27 (t, $J = 7.2$ Hz, 1H), 4.71 (br d, $J = 8.9$ Hz, 1H), 3.85 (br s, 1H), 3.70 (dd, $J = 10.1$, 3.5 Hz, 1H), 3.60 (dd, $J = 10.2$, 4.5 Hz, 1H), 2.05–1.98 (m, 5H), 1.44 (s, 9H), 1.38–1.22 (m, 22H), 0.90–0.86 (m, 12H), 0.04 (s, 6H).

$^{13}$C NMR (100 MHz, CDCl$_3$): δ 169.71, 155.41, 136.77, 124.74, 79.29, 76.68, 76.68, 73.78, 61.79, 53.70, 32.34, 31.91, 29.67, 29.65, 29.59, 29.47, 29.35, 29.22, 28.89, 28.36, 25.82, 22.68, 21.25, 18.21, 14.11, −5.57, −5.59.

MS (APCI, M+H$^+$): $m/z$ found 556.4.

(2S,3R,E)-2-((tert-Butoxycarbonyl)amino)-1-hydroxyoctadec-4-en-3-yl acetate (**3c**)

A solution of HF-pyridine was prepared by addition of ca. 70% HF-pyridine (6.6 mg) to a solution of pyridine (10 µl) in THF (0.13 ml). This solution was added to a solution of **3b** (19 mg, 0.034 mmol) in THF (0.55 ml) at 0 °C. The mixture was stirred for 1 h, and slowly warmed to 50 °C over 1 h, and stirred for 18 h. The reaction was quenched by addition of saturated NaHCO$_3$ solution (30 µl), and diluted with water. The aqueous layer was extracted with ethyl acetate, and combined organic layers were dried over Na$_2$SO$_4$ and filtered, and concentrated in vacuo. The crude material was purified by flash chromatography (10–30% ethyl acetate in hexanes) to yield **3c** (13 mg, 0.029 mmol) as a white solid.

$^1$H NMR (400 MHz, CDCl$_3$): δ 5.81–5.74 (m, 1H), 5.46 (dd, $J = 15.4$, 7.8 Hz, 1H), 5.27 (t, $J = 7.3$ Hz, 1H), 4.96 (br d, $J = 8.5$ Hz, 1H), 3.82–3.73 (m, 1H), 3.65 (d, $J = 3.5$ Hz, 2H), 2.09 (s, 3H), 2.06–2.00 (m, 2H), 1.44 (s, 9H), 1.39–1.23 (m, 22H), 0.88 (t, $J = 7.0$, 3H).

$^{13}$C NMR (100 MHz, CDCl$_3$): δ 170.79, 155.80, 137.18, 124.50, 79.68, 74.48, 61.86, 54.36, 32.28, 31.90, 29.66, 29.65, 29.63, 29.56, 29.43, 29.33, 29.19, 28.82, 28.31, 22.66, 21.21, 14.09.

MS (APCI, M+H$^+$): $m/z$ found 442.4.

(2S,3R,E)-3-Acetoxy-1-hydroxyoctadec-4-en-2-aminium trifluoroacetate (**3-O-AS**)

To a solution of **3c** (18 mg, 0.04 mmol) in DCM (0.56 ml) at 0 °C was added trifluoroacetic acid (240 µl, 3.14 mmol), and triethylsilane (19.3 µl, 0.12 mmol). The mixture was stirred for 2 h, and concentrated in vacuo to yield **3-O-AS** (18 mg, 0.039 mmol) as a pale-yellow oil.

$^1$H NMR (400 MHz, CDCl$_3$): δ 5.91–5.82 (m, 1H), 5.53–5.46 (m, 1H), 5.37 (dd, $J = 15.3$, 6.9 Hz, 1H), 3.95–3.87 (m, 1H), 3.83–3.71 (m, 1H), 3.48 (br s, 1H), 2.11–2.00 (m, 5H), 1.39–1.19 (m, 22H), 0.88 (t, $J = 7.0$, 3H).

$^{13}$C NMR (100 MHz, CDCl$_3$): δ 170.93, 139.50, 121.36, 71.82, 58.05, 56.05, 32.23, 31.93, 29.71, 29.68, 29.67, 29.67, 29.59, 29.41, 29.36, 29.24, 28.54, 22.69, 20.76, 14.11.

HRMS (DART, M+H$^+$): $m/z$ calcd. for C$_{20}$H$_{40}$NO$_3^+$ 342.3003, found 342.3002.

N-((2S,3R,E)-1,3-Dihydroxyoctadec-4-en-2-yl)acetamide (**N-AS**)

**Synthesis of N-AS from 1-O-AS**

To a solution of **1-O-AS** (13.5 mg, 0.029 mmol) in DCM (0.53 ml) at RT. was added triethylamine (90 µl, 0.64 mmol), and was stirred for 3 h. The mixture was concentrated in vacuo, and purified by flash chromatography (1–5% methanol in DCM) to yield **N-AS** (9 mg, 0.026 mmol) as a white solid.

**Synthesis of N-AS from 3-O-AS**

To a solution of **3-O-AS** (17 mg, 0.037 mmol) in DCM (0.6 ml) at RT was added triethylamine (115 µl, 0.82 mmol) and was stirred for 3 h. The mixture was concentrated in vacuo, and purified by flash chromatography (1–5% methanol in DCM) to yield **N-AS** (10 mg, 0.029 mmol) as a white solid.

$^1$H NMR (400 MHz, CDCl$_3$): δ 6.33 (br d, $J = 5.5$ Hz, 1H), 5.83–5.75 (m, 1H), 5.53 (dd, $J = 15.4$, 6.4 Hz, 1H), 4.35–4.31 (m, 1H), 3.99–3.89 (m, 2H), 3.71 (dd, $J = 11.0$, 3.1 Hz, 1H), 2.09–2.02 (m, 5H), 1.40–1.22 (m, 22H), 0.88 (t, $J = 7.0$, 3H).

$^{13}$C NMR (100 MHz, CDCl$_3$): δ 171.09, 134.18, 128.59, 74.24, 62.12, 54.56, 32.28, 31.90, 29.66, 29.63, 29.61, 29.48, 29.33, 29.22, 29.13, 23.31, 22.66, 14.09.

HRMS (DART, M+H$^+$): $m/z$ calcd. for C$_{20}$H$_{40}$NO$_3^+$ 342.3003, found 342.3031.

**LC-ELSD analysis.** To a solution of 1-O-AS (0.25 mg) or 3-O-AS (0.25 mg) in water (400 µl) and DMSO (50 µl) at RT was added sodium phosphate buffer (pH 7.4, 0.1 M, 50 µl). The solution was mixed homogeneously with micropipette and was incubated at 25 °C for 10 min. Reverse-phase HPLC was performed using Agilent 1220 coupled with Agilent 1260 Infinity ELSD. Phenomenex Kinetex 1.7 µm C8 100 A (50 × 21 mm LC column) at a flow rate of 0.5 ml min$^{-1}$ was used. A gradient of acetonitrile with 0.1% formic acid was used as an eluent.

**LC-MS/MS analysis for structure elucidation of N-AS.** To determine whether N-AS was synthesized by acetyl-CoA and sphingosine within SphK1, 10 mM acetyl-CoA (Sigma-Aldrich, A2056) and 100 µM sphingosine (Sigma-Aldrich, S7049) were incubated with purified SphK1 (Cayman, 10348) at 37 °C for 24 h in acetylation buffer (50 mM Tris-HCl pH 7.6, 1 mM DTT, 1 mM EDTA, and 10% glycerol, all from Sigma-Aldrich). Also, SphK1 WT, SphK1 D178A, and SphK1 F192A were incubated with 100 µM sphingosine and acetyl-CoA ranging from 0 to 10 mM. Reactions were quenched by adding equal volume of cold methanol. A 100 µl aliquot of the reaction mixture was mixed with 30 µl of distilled water containing 5 ng ml$^{-1}$ of furosemide (internal standard) for 1 min and vigorously mixed with 0.7 ml$^{-1}$ of ethyl acetate for 20 min. After centrifugation at 16,000×$g$ for 10 min, the mixtures were kept for 2 h for freezing aqueous phase. The upper ethyl acetate layer was transferred to a clean tube and evaporated to dryness under a gentle stream of nitrogen. Then, the dried extract was reconstituted in 100 µl of mobile phase and 10 µl aliquot of the solution was injected into an Agilent 6470 Triple Quad LC–MS/MS system (Agilent, Wilmington, DE) coupled with an Agilent 1260 HPLC system. Separation was performed on a Synergy Polar RP column (4 µm particle size, 2.0 mm × 150 mm, Phenomenex) using a mobile phase that consisted of methanol and water (95:5, v/v) with 0.1% formic acid at a flow rate of 0.2 ml min$^{-1}$. Quantification was carried out using multiple reaction monitoring (MRM) at $m/z$ 340.3→263.3 for N-AS and $m/z$ 328.9→284.8 for furosemide (internal standard) in negative ionization mode and collision energy of 20 eV. The lower limit of quantification was determined to be 0.5 ng ml$^{-1}$ and linearity was observed in the standard range of 0.5–200 ng ml$^{-1}$. Also, to confirm N-AS generation in neurons, microglia, and astrocytes, these cells were treated with Aβ, and sonicated in neuronal and glial culture medium. The supernatants were supplemented with acetyl-CoA (100 µM, Sigma-Aldrich, A2056), SphK1 (20 ng, Cayman, 10348), or sphingosine (1 µM, Sigma-Aldrich, S7049), and incubated at 37 °C for 2 h. To evaluate N-AS in microglia and neurons in vivo, the cells were isolated from WT mice, APP/PS1 mice and APP/PS1 mice treated with N-AS at 9 months of age. The cells were sonicated and resuspended in glial and neuronal culture medium, respectively. A 100 µl aliquot of each cell lysate were used for the determination of N-AS using a method previously described.

**LC-MS/MS analysis of the acetylated site in COX2.** To identify the acetylation site of COX2, N-AS (Sigma-Aldrich, 01912)-treated COX2 enzymes (LSBio, LS-G21094) were immediately precipitated with trichroloacetic acid (Merck) and dried. The dried extract was resuspended in 10 µl of 5 M Urea solution and

incubated with 1 μg sequencing-grade modified porcine trypsin (Promega) in 0.1 M ammonium bicarbonate buffer at 37 °C for 16 h. The sample was then treated with 1 M DTT (GE Healthcare) at RT for 1 h, followed by alkylation with 1 M iodoacetamide (Sigma-Aldrich) for 1 h. For sequencing, the protein samples were loaded onto a ZORBAX 300SB-C18 column (3.5 μm, 1.0 mm i.d. × 150 mm, Agilent). The column was placed in-line with an UltiMate 3000 system (Dionex, USA) and a splitter system was used to achieve a flow rate of 100 μl min$^{-1}$. Analytes were eluted with a mobile phase consisting of water: formic acid (100:0.2 v/v) (phase A) and acetonitrile: formic acid (100:0.2 v/v) (phase B) in gradient elution mode for 77 min. Eluted peptides were directly electrosprayed into and MicroQ-TOF III mass spectrometer (Bruker Daltonics, 255748, Germany) by applying 4.5 kV of capillary voltage and a normalized collision energy of 7 eV. The peptides were verified with BioTools 3.2 SR5 (Bruker Daltonics).

**Peptide and antibodies generation.** Polyclonal and monoclonal anti-acetyl COX2 S565 antibodies were generated using the COX2 peptide including acetylated S565 residue to immunize rabbits and mice (Genscrpit). Double affinity purification was performed using native and acetylated peptides sequentially using Sulfolink columns (Thermo Fisher Scientific).

**Biosynthesis and lipid mediators metabololipidomics.** Lipid mediators metabololipidomics were performed according to previously described procedures with minor modifications[4,30]. To confirm the biosynthesis of lipid mediators by recombinant COX2 treated with N-AS or aspirin, 10 U of human recombinant COX2 (Cayman, 60122) was incubated in the presence of 500 μM aspirin (Sigma-Aldrich, A5376) or N-AS (Sigma-Aldrich, 01912) at 37 °C for 30 min. The enzyme was filtered with a Microcon column (Millipore) to remove unreacted aspirin or N-AS and was then resuspended in 0.1 M Tris-HCl (pH 8.0). Next, 500 μM phenol (Sigma-Aldrich) and 1 μM hematin-porcine (Sigma-Aldrich) were added immediately before incubation with 100 μM AA, EPA or DHA (Cayman) at 37 °C for 2 h, respectively. The mixtures were incubated with 30 U of human 5-LOX (Cayman, 60402) containing 2 mM CaCl$_2$ (Sigma-Aldrich), 1 mM ATP (Amresco) and 5 μM 13-HpODE (Cayman) as activators. Also, to validate the importance of residue S565 for N-AS-acetylation of COX2 and production of SPMs, 10 U of the COX2 S565A mutant was used in the above-described method. In addition, to confirm SPM generation in neurons and microglia, these cells were treated with Aβ and sonicated in neuronal and glial culture medium. The supernatants were supplemented with acetyl-CoA (100 μM, Sigma-Aldrich, A2056), SphK1 (20 ng, Cayman, 10348), or sphingosine (1 μM, Sigma-Aldrich, S7049), and incubated at 37 °C for 2 h. To evaluate the relationship of N-AS and SPM production in microglia, microglia were isolated from WT, APP/PS1, and APP/PS1 treated with N-AS mice at 9 months of age. The microglia were sonicated and resuspended in microglial culture medium. The supernatants were incubated with 100 μM AA, EPA, and DHA (Cayman) at 37 °C for 2 h. A 50 μl aliquot of the mixture from each cell lysate was added to a 250 μl aliquot of 1 ng ml$^{-1}$ of 15S-LXA$_4$-d5 (internal standard, Cayman) methanol solution. After vortexing for 10 min and centrifuging at 15,974×$g$ for 5 min, the supernatant was subjected to the analysis of the lipid mediators using an Agilent 6470 Triple Quad LC-MS/MS system (Agilent) coupled to an Agilent 1290 HPLC system. Chromatographic separation was achieved using a Synergi Polar RP (4 μm, 2.0 mm i.d. × 150 mm, Phenomenex). Analytes were eluted with a mobile phase consisting of water: formic acid (100:0.1 v/v) (phase A) and methanol: formic acid (100:0.1 v/v) (phase B) in gradient elution mode for 23 min a flow rate of 0.2 ml min$^{-1}$. The elution gradient was as follows: from 0 to 0.1 min the content of phase A was 37% and the content of phase A was decreased to 10% and maintained for 5.5 min. The content of phase A was increased back to 37% at 6.0 min and maintained for 23 min. Quantification was carried out using MRM in negative ionization mode and collision energy of 10–15 eV. The mass transition from Q1 to Q3 is shown in Supplementary Table 2. In this study, the peak areas for all components were automatically integrated using MassHunter B 06.00.

**Cell culture.** To confirm the generation and roles of N-AS in neurons, cells from E18 C57BL/6 mice were prepared as previously described[11]. Cortices were dissected and then dissociated followed by incubation in papain (Worthington) at 37 °C for 15 min. Neurons were plated on poly-L-lysine-coated coverslips with neuronal culture medium, serum-free Neurobasal medium (Gibco) containing 2% B27 supplements (Gibco), 1 mM Glutamax supplement (Gibco), and 100 U ml$^{-1}$ streptomycin/ 100 U ml$^{-1}$ penicillin (Gibco) at 37 °C in a humidified atmosphere of 5% CO$_2$. Aβ 1–42 (10 μM, 24 h, Invitrogen, 03112) was exposed to neurons and the cells were analyzed for the generation and roles of N-AS. Primary astrocyte cultures were prepared from C57BL/6 mice as described previously[11]. In brief, after removal of the meninges, postnatal day 7 (P7) mouse brain tissues were minced and incubated in a rocking water bath at 37 °C for 30 min in the presence of 0.25% trypsin-EDTA (Sigma-Aldrich). Enzyme-digested dissociated cells were triturated with astrocyte-specific medium (DMEM/F12 containing 10% FBS and 1% penicillin-streptomycin) and centrifuged at 155×$g$ for 8 min. The pellet was resuspended in astrocyte-specific medium and passed through a 40-μm cell strainer. The filtrate was allowed pre-adherence for 30 min to remove any contamination from fibroblasts before being seeded in dishes and the addition of astrocyte-specific

medium. To split the cells, Ara-C was added to the dishes and they were then placed in a heated shaker for 6–7 h. The medium was removed from the dishes, and trypsin-EDTA was added to the dish and incubated at 37 °C for 5–10 min. After centrifugation at 155×$g$ for 8 min, the supernatant was removed, and the astrocytes were maintained in culture by feeding every 1–2 weeks with astrocyte-specific medium. Primary microglia cultures were prepared from C57BL/6 mice as described previously[11]. In brief, the cortices of 2-months-old mice were minced in Hibernate A (Gibco)/B27 supplements (Gibco) medium and dissociated using papain (Worthington) solution. After tissue trituration, cells were separated by Optiprep (Sigma-Aldrich) density gradient centrifugation. Microglia were plated with microglia-specific medium (DMEM/F12 containing 10% FBS and 1% penicillin-streptomycin). The immortalized human microglia (Applied Biologics Materials, T0251) were cultured in PriGrow III (Applied Biologics Materials, TM003) (supplemented with 10% FBS and 1% penicillin-streptomycin) at 37 °C in a humidified atmosphere of 5% CO$_2$. In order to assess the clinical importance of N-AS in AD microglia, Aβ 1–42 (50 μM, 24 h, Invitrogen, 03112) was exposed to human microglia before treated with zileuton (50 μM, 24 h, Sigma-Aldrich, Z4277). The human microglia treated with Aβ 1–42 and/or zileuton was incubated with N-AS (2.5 μM, 2 h, Sigma-Aldrich, 01912), and were analyzed for the generation and roles of N-AS.

**Cell isolation.** Adult neurons and microglia were isolated from the mice brain as previously described[11]. In brief, the cortex of WT (1, 3, 5, or 9-months-old), APP/PS1 (1, 3, 5, or 9-months-old), APP/PS1 treated with N-AS (9-months-old) or APP/PS1 treated with aspirin (9-months-old) mice were minced in Hibernate A (Gibco)/B27 (Invitrogen) medium and dissociated using papain (Worthington) solution. After tissue trituration, cells were separated by Optiprep (Sigma-Aldrich) density gradient centrifugation. The purity of the fractionated neurons and microglia was acutely confirmed by neuronal (NeuN, mouse, 1:200, Millipore, MAB377) and microglial (Iba1, rabbit, 1:1000, Wako, 019-19941) markers and the isolated neurons and microglia were acutely analyzed for the acetylation assay and lipid mediators metabololipidomics.

**Histological analysis.** Thioflavin S (Sigma-Aldrich) staining was carried out according to previously described procedures[31]. We used 6E10 (mouse, 1:100, Signet, SIG39300), anti-20G10 (mouse, 1:1000, provided by D. R. Howlett, GlaxoSmithKline, Harlow, Essex, UK) for Aβ42, anti-G30 (rabbit, 1:1000, provided by D. R. Howlett) for Aβ40, SMA (mouse, 1:400, Sigma-Aldrich, A2547), AT8 (mouse, 1:500, Thermo Fisher Scientific, MN1020), Caspase-3 (mouse, 1:200, Novus Biologicals, NB100-56708), ac-S565 (rabbit, 1:100), COX2 (mouse, 1:250, Thermo Fisher Scientific, 35-8200), Iba1 (rabbit, 1:500, Wako, 019-19941 and goat, Abcam, ab5076), NeuN (mouse, 1:500, Millipore, MAB177), GFAP (rabbit, 1:500, Dako, N1506 and chicken, 1:500, Abcam, ab4674), and Lamp1 (mouse, 1:200, Abcam, ab24170). The sections were analyzed with a laser-scanning confocal microscope (FV3000; Olympus) or with a BX51 microscope (Olympus). Meta-Morph software (Molecular Devices) was used to quantification. Three-dimensional reconstruction of microglia was recorded and analyzed using IMARIS software (Bitplane)[11]. For the analysis of age-dependent microglial COX2$^+$ CX3CR1$^+$ cells in WT and APP/PS1 mice, COX2 (rabbit, 1:100, Abcam, ab15191) and CX3CR1 (mouse, 1:100, BioLegend, 149008) antibody were used and the images were acquired on Alexa 488 and APC Channel with performed using Operetta CLS and Analysis Software 4.5 (Perkin-Elmer).

**Western blotting.** Samples were lysed in RIPA buffer (Cell signaling Technologies), then subjected to SDS-PAGE and transferred to a nitrocellulose membrane. Membranes were blocked with 5% milk, incubated with primary antibody and then incubated with the appropriate horseradish peroxidase-conjugated secondary antibody[31]. Primary antibodies to the following proteins were used: ac-S565 (rabbit, 1:500 and mouse, 1:500), COX2 (rabbit, 1:1,000, Abcam, ab15191), Synaptophysin (rabbit, 1:2000, Abcam, ab32127), MAP2 (chicken, 1:10,000, Abcam, ab5392), Synapsin 1 (rabbit, 1:1000, Synaptic systems, 106 103), PSD65 (mouse, 1:1000, Millipore, MAB1596), and β-actin (mouse, 1:1,000, Santa Cruz, SC-47778). We performed densitometric quantification using the ImageJ software (National Institutes of Health). Images have been cropped for presentation.

**Lipid extraction and sphingosine quantification.** To quantify the sphingosine levels, the dried lipid extract was resuspended in 0.2% Igepal CA-630. Four microliters of the lipid extracts was added into 20 μl of NDA derivatization reaction mixture (25 mM borate buffer, pH 9.0, containing 2.5 mM each of NDA and NaCN). The reaction mixture was diluted 1:3 with ethanol, incubated at 50 °C for 10 min and centrifuged at 15,493×$g$ for 5 min. A 30 μl aliquot of the supernatant was then transferred to a sampling glass vial and 5 μl was applied onto a UPLC system for analysis. The fluorescent sphingosine was monitored using a model 474 scanning fluorescence detector (Waters). Quantification of the sphingosine was calculated from sphingosine calibration curves using the Waters Millennium software.

**ELISA.** For measurement of acetyl-CoA in neurons and microglia, we used commercially available ELISA kits (Abcam, ab87546) according to the manufacturer's

instructions. For measurement of Aβ40 and Aβ42, we used commercially available ELISA kits (Invitrogen, KHB3481, and KHB3441). Cortex of mice was homogenized in buffer containing 0.02 M guanidine. ELISA was then performed for Aβ40 and Aβ42 according to the manufacturer's instructions.

**Microsomal stability**. An N-AS stock solution (10 mM dissolved in methanol) was diluted with methanol to make a concentration of 100 μM. A 10 μl aliquot of this solution was then diluted with 90 μl of 100 mM potassium phosphate buffer (pH 7.4). 10 μl of this N-AS solution was incubated at 37 °C with human or mouse liver microsomes (80 μl of 0.625 mg protein ml$^{-1}$ phosphate buffer) for 10 min before adding of 10 μl of an NADPH regenerating system to start the reaction. Reactions were stopped at 0, 15, 30, 60, and 90 min time-points by the addition of 200 μl cold acetonitrile (containing propranolol as internal standard at 20 ng ml$^{-1}$) and centrifuged at 17,968×$g$ for 10 min. A 2 μl aliquot of the supernatant was injected into the LC-MS/MS system. The ratio of peak area of test compound/internal standard was used to determine the % remaining of N-AS over time. The half-life (T1/2) and % compound remaining after 30 min were then calculated.

**Cytochrome P450 inhibition**. N-AS stock solution (10 mM dissolved in methanol) was serially diluted with methanol to make concentrations of 100, 250, 500, 2500, and 10,000 μM. A 10 μl aliquot of this solution was then diluted using 90 μl of 100 mM potassium phosphate buffer (pH 7.4). Probe substrate cocktail solutions for seven major cytochrome P450 isozymes (phenacetin (100 μM for 1A2), bupropion (50 μM for 2B6), diclofenac (20 μM for 2C9), mephenytoin (100 μM for 2C19), dextromethorphan (5 μM for 2D6), chlorzoxazone (50 μM for 2E1), and midazolam (5 μM for 3A4)) were prepared with phosphate buffer. A 10 μl aliquot of N-AS solution and a 10 μl aliquot of each probe substrate cocktail solution was incubated at 37 °C with human liver microsomes (80 μl of 0.625 mg protein ml$^{-1}$ phosphate buffer) for 10 min before the addition of 10 μl of the NADPH regenerating system to start the reaction. Reactions were stopped at 15 min by the addition of 200 μl cold acetonitrile (containing propranolol as internal standard at 20 ng ml$^{-1}$) and centrifuged at 17,968×$g$ for 10 min. A 2 μl aliquot of the supernatant was injected into the LC-MS/MS system. The ratio of peak area of test compound /internal standard was used to determine the % metabolic activity of probe substrate according to the concentrations of N-AS added.

**PK and brain distribution of N-AS**. C57BL/6 mice were administered N-AS (Sigma-Aldrich, 01912) at doses of 1 mg kg$^{-1}$ (i.v.) or 10 mg kg$^{-1}$ (p.o.). Venous blood samples were collected at 5, 15, and 30 min, and 1, 2, 4, 8, and 24 h postdose. A 30 μl of plasma was separated from whole blood by centrifugation and stored at −80 °C until analysis. Also, brain samples were collected at 1, 2, 4, and 24 h postdose and heart, liver, and kidney samples were collected at 24 h postdose, thoroughly rinsed with physiological saline and weighed. Twenty percentage of tissue homogenates were prepared by homogenizing the tissue samples with four volumes of saline. A 30 μl aliquot of plasma or a 50 μl aliquot of tissue homogenates were deproteinized by addition of 200 μl cold methanol (containing naringenin as internal standard at 20 ng ml$^{-1}$) and centrifuged at 15,974×$g$ for 10 min. A 10 μl aliquot of the supernatant was injected into Agilent 6470 LC-MS/MS system. Separation was performed on a Synergy Polar RP column (4 μm particle size, 2.0 mm ×150 mm, Phenomenex) using a mobile phase that consisted of methanol and water (85:15, v/v) with 0.1% formic acid at a flow rate of 0.2 ml min$^{-1}$. Quantification was carried out using multiple reaction monitoring (MRM) at $m/z$ 340.3→263.3 for N-AS and $m/z$ 271.1→151.3 for naringenin (internal standard) in negative ionization mode and collision energy of 20 eV. $p$, and relative oral BA values were calculated as follows: BA% = AUC$_{p.o.}$ normalized by p.o. dose/AUC$_{i.v.}$ normalized by i.v. dose × 100.

**Mice**. Transgenic mouse lines over-expressing the hAPP695swe (APP) and presenilin-1M146V (PS1) mutations were originated from GlaxoSmithKline (Harlow, UK)[32] and maintained as described previously[31]. We purchased 5×FAD mice from Jackson Labs (Stock number. 34840-JAX). Because APP/PS1 and 5×FAD mice show sex differences in disease progression, we used only male mice. To examine possible therapeutic effect of N-AS and aspirin in APP/PS1 mice, the N-AS (30 mg kg$^{-1}$, Sigma-Aldrich, 01912) was injected daily s.c. for 8 weeks to 7-months-old APP/PS1 mice until the age of 9 months, and aspirin (2 mg kg$^{-1}$, Sigma-Aldrich, A5376)[33] was administrated orally to 7-months-old APP/PS1 mice for 8 weeks. Also, to confirm possible therapeutic efficacy of N-AS in 5×FAD, the N-AS (30 mg kg$^{-1}$, Sigma-Aldrich, 01912) was injected daily s.c. for 4 weeks to 3-months-old 5×FAD mice until the age of 4 months. Block randomization method was used to allocate the animals to experimental groups. To eliminate the bias, we were blinded in experimental progress such as data collection and data analysis. Mice were housed at temperature of 22 °C and a 12 h day/12 h night cycle with free access to tap water and food pellets. All protocols were approved by the Kyungpook National University Institutional Animal Care and Use Committee.

**Flow cytometry**. Brain microglia polarity and immune cells in brain and blood were analyzed by flow cytometry. Cells from WT, APP/PS1, and APP/PS1 treated with N-AS mice brain were prepared as previously described with minor

modifications[34]. Brains of WT, APP/PS1, and APP/PS1 treated with N-AS mice were dissected and immediately transferred in ice-cold HBSS (Gibco). After gentle mincing, the brain was digested in a HBSS solution containing collagenase P (0.2 mg ml$^{-1}$, Roche), dispase II (0.8 mg ml$^{-1}$, Roche), DNase I (0.01 mg ml$^{-1}$, Roche), and collagenase A (0.3 mg ml$^{-1}$, Roche) at 37 °C for 1 h under gentle rocking. Digestion was stopped by adding FBS (Gibco) on ice. The supernatants were centrifuged at 250×$g$ for 10 min at 4 °C. The pellet was resuspended in 25% BSA (Gibco)/PBS (Gibco) for myelin removal. Following a centrifugation step at 3000×$g$ for 30 min at 4 °C, the myelin containing supernatant was discarded. The cell pellets were then resuspended in 1 ml of HBSS and filtered through a 40 μm mesh, followed by a washing step in HBSS. The cell pellets were resuspended in 1 ml of red blood cell lysis buffer (BD Biosciences) and incubated at RT for 10 min for lysis of erythrocytes. Subsequently, 2 ml of MACs buffer (Miltenyl Biotec, 130-091-222) was added and centrifuged at 250×$g$ for 10 min at 4 °C. Cells from WT, APP/PS1, and APP/PS1 treated with N-AS mice blood were prepared as previously described with minor modifications[35]. Blood was collected in sodium-heparin tube (BD, 367871) by cardiac puncture. Red blood cells were lysed once at 4 °C for 10 min in 0.15 M NH$_4$Cl (STEMCELL Technologies), washed once with PBS (Gibco), and counted using cell counter (Logos biosystems). The cells were stained with the following antibodies for brain microglia polarity[36,37], macrophage subsets[34,38], monocytes[38,39], neutrophils[38,40], T cells[41], and B cells[42]: mouse anti-CD45 PerCp Cy5.5 (550994; BD Bioscience), mouse anti-CD11b PE (557397; BD Bioscience), mouse anti-CD86 PE-Cy7 (560582; BD Bioscience), mouse anti-CD206 APC (141707; Bio legend), mouse anti-CD45 APC-Cy7 (557659; BD Bioscience), mouse anti-Gr-1 FITC (553126; BD Bioscience), mouse anti-MHCII PE (557000; BD Bioscience), mouse anti-F4/80 APC (14-4801-82, Thermo Fisher Scientific), mouse anti-CD206 PE-Cy7 (25-2061-82; Thermo Fisher Scientific), mouse anti-Ly6C FITC (553104; BD Bioscience), mouse anti-Ly6G APC-Cy7 (557661; BD Bioscience), mouse anti-CD11b APC (553312; BD Bioscience), mouse anti-CD115 PE (12-1152-82; Thermo Fisher Scientific), mouse anti-CD4 FITC (553047; BD Bioscience), mouse anti-B220 PE (50-0452; Tonbo Bioscience), mouse anti-lineage biotin (130-090-858, Miltenyl Biotec), and anti-biotin streptavidin PB (S11222; Invitrogen). After an incubation period of 30 min at 4 °C, fluorescence data were acquired on a BD FACSAriaIII flow cytometer (BD) using Diva software (BD) and further analyzed using FlowJo analytical software (Tree Star, Inc.).

**RNAseq**. Total RNA was isolated in microglia derived from WT, APP/PS1, and APP/PS1 mice treated with N-AS using Trizol reagent (Invitrogen). RNA quality was assessed by Agilent 2100 bioanalyzer using the RNA 6000 Nano Chip (Agilent Technologies, Amstelveen, The Netherlands), and RNA quantification was performed using ND-2000 Spectrophotometer (Thermo Inc., DE, USA). To prepare and sequence libraries, libraries were prepared from 1 μg of total RNA using the SMARTer Stranded RNA-Seq Kit (Clontech Laboratories, Inc., USA). rRNA was removed using RIBO COP rRNA depletion kit (LEXOGEN, Inc., Austria). The rRNA depleted RNAs were used for the cDNA synthesis and shearing, following manufacture's instruction. Indexing was performed using the Illumina indexes 1–12. The enrichment step was carried out using of PCR. Subsequently, libraries were checked using the Agilent 2100 bioanalyzer (DNA High Sensitivity Kit) to evaluate the mean fragment size. Quantification was performed using the library quantification kit using a StepOne Real-Time PCR System (Life Technologies, Inc., USA). High-throughput sequencing was performed as paired-end 100 sequencing using HiSeq 2500 (Illumina, Inc., USA). Then, total RNA-Seq reads were mapped using TopHat software tool in order to obtain bam file (alignment file). Read counts mapped on transcripts region were extracted from the alignment file using bedtools and Bioconductor that uses R statistical programming language. The alignment file also was used for assembling transcripts, estimating their abundances and detecting differential expression of genes, linc RNAs or isoforms. And we used the FPKM (fragments per kilobase of exon per million fragments) as the method of determining the expression level of the gene regions. Quantile normalization method was used for comparison between samples. Functional gene classification was performed by DAVID (http://david.abcc.ncifcrf.gov/).

**Phagocytosis assay**. The phagocytic activity of adult microglia in the cortex of live brain slices was analyzed as following protocol[11]. Brains from WT, APP/PS1, and APP/PS1 mice treated with N-AS were washed in carbogen-saturated (95% O$_2$ and 5% CO$_2$) artificial cerebrospinal fluid (ACSF) containing (in mM): NaCl 126; KCl 2.5; MgSO$_4$ 1.3; CaCl$_2$ 2.5; NaH$_2$PO$_4$ 1.25; NaHCO$_3$ 26; and D-glucose 10; pH 7.4 (all from Sigma-Aldrich). Coronal slices (130 μm) were prepared using a vibratome (Leica Biosystems) at 4 °C and allowed to rest in ACSF buffer at RT for 1 h before incubation with fluorescent carboxylated microspheres (1 μm diameter, Fluo-Spheres, Invitrogen, 1:2000) in PBS (Gibco) containing 4.5 g l$^{-1}$ D-glucose (Sigma-Aldrich) at 37 °C for 1 h. The slices were washed and fixed with 4% PFA. To visualize microglia, slices were permeabilized (2% Triton-X100, 2% BSA and 10% normal goat serum in PBS) and incubated with Iba1 (rabbit, 1:500, Wako), followed by donkey anti-rabbit Alexa 594 (1:1000, Invitrogen). The sections were analyzed with a laser-scanning confocal microscope (FV3000; Olympus). Meta-Morph software (Molecular Devices) was used to quantification.

**RNA isolation and real-time PCR analysis**. For total RNA extraction from the microglia derived from WT, APP/PS1, APP/PS1 mice treated with N-AS, and APP/PS1 mice treated with aspirin, the cells were isolated from the cortex of WT, APP/PS1, APP/PS1 mice treated with N-AS, or aspirin using the methods described above. Total RNA was isolated using the kit components according to the manufacturer's instructions (RNeasy mini kit, 74134, QIAGEN). cDNA was synthesized from 5 μg of total RNA using a commercially available kit (Takara). Quantitative real-time PCR was performed using a Corbett research RG-6000 real-time PCR instrument. Primers are described in Supplementary Table 5.

**Behavioral studies**. We performed behavioral studies to assess spatial learning and memory in the Morris water maze as previously described[11,31]. Animals were given four trials per day for 10 days to learn the task. At day 11, animals were given a probe trial in which the platform was removed. Fear conditioning was conducted by previously described techniques[32]. On the conditioning day, mice were individually placed into the conditioning chamber. After a 60 s exploratory period, a tone (10 kHz, 70 dB) was delivered for 10 s; this served as the conditioned stimulus (CS). The CS co-terminated with the unconditioned stimulus (US), a scrambled electrical footshock (0.3 mA, 1 s). The CS-US pairing was delivered twice at a 20 s intertrial interval. On day 2, each mouse was placed in the fear-conditioning chamber containing the same exact context, but with no administration of a CS or foot shock. Freezing was analyzed for 5 min. On day 3, a mouse was placed in a test chamber that was different from the conditioning chamber. After a 60 s exploratory period, the tone was presented for 60 s without the footshock. The rate of freezing response of mice was used to measure the fear memory. The open field test was used for locomotion and anxious behaviors as previously described[11]. The open field box consisted of a square box. Each animal was placed in the box for 10 min. Overall activity in the box was measured, and the amount of time and distance traveled in the center arena was noted. After each trial, the test chambers were cleaned with a damp towel and distilled water followed by 70% alcohol. The light-dark test was used for assessing the anxiety-like behavior as previously described[43]. One chamber was brightly illuminated, whereas the other chamber was dark. Mice were placed into the dark chamber and allowed to move freely between the two chambers with the door open for 10 min. The total number of transitions, latency to first enter the light chamber, distance traveled, and time spent in each chamber were recorded.

**Live-cell phagocytosis assay**. Live-cell phagocytosis assays were performed according to previously described procedures with minor modifications[44]. Briefly, in a glass bottom cell culture dish (SPL, 101350), adult SV40 immortalized human microglia cells (Applied Biologics Materials, T0251) were seeded for phagocytosis assay. The seeded cells were incubated with 0 or 50 μM Aβ (Invitrogen, 03112) at 37 °C for 24 h, before treated with 0 or 50 μM zileuton (Sigma-Aldrich, Z4277) at 37 °C for 24 h. After incubation, the microglia were treated with 0 or 2.5 μM N-AS (Sigma-Aldrich, 01912) at 37 °C for 2 h. FITC beads (Invitrogen) preopsonized in FBS (Gibco) were added at a concentration at 0.01% and then immediately observed using a laser scanning confocal microscope equipped with a live cell chamber system (FV3000, Olympus) for 2 h. Confocal images were collected every 5 min for 2 h and analyzed using the ImageJ software (National Institutes of Health).

**Statistical analysis**. Sample sizes were determined by G-Power software (with $\alpha = 0.05$ and power of 0.8). In general, statistical methods were not used to re-calculate or predetermine sample sizes. Variance was similar within comparable experimental groups. Experimenters were blinded to the identity of experimental groups until the end of data collection and analysis for at least one of the independent experiments. Comparisons between two groups were performed with Student's t-test. In cases where more than two groups were compared to each other, a one way analysis of variance (ANOVA) was used, followed by Tukey's HSD test. All statistical analyses were performed using SPSS statistical software and Graph-Pad Prism 7.0 software. *$P < 0.05$, **$P < 0.01$, ***$P < 0.001$, and ****$P < 0.0001$ were considered to be significant.

**Reporting summary**. Further information on research design is available in the Nature Research Reporting Summary linked to this article.

## Data availability

The datasets generated and/or analyzed during the current study are also available from the corresponding authors upon reasonable request. The source data underlying Figs. 1a, b, 1h, 2c, 3a–i, 4b–e, 5a–g, 6a–p and Supplementary Figs. 1i, 3g, 4b–d, 4g, 5a–i, 6a–d, 6f, 7a–h, 7j–m, 7o–q, and 8b–i are provided as a Source Data file. RNAseq data is deposited in the GEO database (GSE146133) https://www.ncbi.nlm.nih.gov/geo/query/acc.cgi?acc=GSE146133.

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

## Acknowledgements

This research was supported by the Basic Science Research Program (2017R1A4A1015652, 2020R1A2C3006875) and the Original Technology Research Program for Brain Science (2018M3C7A1056513) of the NRF funded by the Korean government, MSIT. This research was also supported by a grant of the Korea Health Technology R&D Project through the KHIDI, funded by the Ministry of Health & Welfare, Republic of Korea (HI16C2131) and Research Fund (1.180002.01) of UNIST.

## Author contributions

J.Y.L. and S.H.H. designed and performed experiments and wrote the paper. M.H.P., I.S.S., and M.K.C. performed LC-MS/MS experiments and analyzed data. E.Y. and C.M.P. synthesized and provided ASs. I.S.S., C.M.P., H.J.K., S.H.K., E.H.S., H.K.J., and J.S.B. interpreted the data and reviewed the paper. H.K.J. and J.S.B. designed the study and wrote the paper. All authors discussed results and commented on the manuscript.

## Competing interests

The authors declare no competing interests.
