## [Peer Review File · Nature Communications]

Reviewers' comments:

Reviewer #1 (Remarks to the Author):

The authors previously described in their recent published paper in Nature Comm 2018 the role of SphK1 in neurons in AD pathogenesis using animal models and human AD cells with a special focus on the impact on microglia mediated functions. They described the mechanism by which SphK1 leads to secretion of SPMs which involves the acetylation on serine 565 of COX2. The resulting increase of phagocytic microglia due to the release of proresolving lipids from neurons. In this former paper they reported that the enzyme SphK1 level and activity is decreased in AD APP/PS1 mice specifically in neurons but not in microglia or astrocytes, and that the neuronal enzyme activity, but not that in glial cells, affected the AD pathology and the cognitive deficit. Collectively, these results suggested and supported that increased neuronal SphK1 activity in APP/PS1 mice reduced Abeta load and improved cognitive deficits. They also reported that the increased neuronal SphK1 improve the neuroinflammation in AD brains. Overall this 2018 paper describes SphK1 dysfunction in neurons as the driving mechanism for microglia pathology and cognitive deficits in AD brain.

In this new paper, the authors reproduce faithfully the experimental design of the previous study of 2018 but focus their study on microglia. Moreover, they study more in-depth how SphK1 leads to acetylation of COX2 expanding the previous study of 2018 and including investigation on the involvement of N-acetyl sphingosine (N-AS). In this respect, the present study although interesting is incremental rather than proposing a new hypothesis for AD pathology.

The novel findings report that N-AS is reduced in microglia and neurons in AD, resulting in reduced SPMs release and the authors provide evidence for differentiating the molecular mechanism leading to reduced SPMs release and operating in microglia from the one operating in neurons. This was done in vitro by exposing microglia or neurons to Abeta.

The amelioration of AD pathology and behavioral outcomes were tested in AD mouse models treated with N-AS and they reproduced therapeutic effects similar to those previously reported in 2018 paper in mice with increased activation of SphK1 and COX2 acetylation in neurons. Findings in the present paper were additionally confirmed using a second AD mouse model (5xFAD).

Overall the study is certainly interesting but is rather incrementing the knowledge related to the original findings of the 2018 paper than establishing a new concept.

I consider the numerosity of samples in various experiments very low (n=3-4) for a sound statistical analysis of the data (see Figures 3-6). All or most of the vivo data in AD mouse models are in the Suppl information while the relevant ones showing the therapeutic effects of N-AS should appear in the main text.

Reviewer #2 (Remarks to the Author):

This is an interesting paper, expressing the aim to further analyse the role of sphingosine kinase 1 (SphK1) in the acetyl-CoA-dependent acetylation of COX-2, and the mechanisms involved of this in Alzheimer's disease (AD). The main claims of the paper are that: 1) acetyl-CoA binds the ATP-binding site in SphK1; 2) reaction of acetyl-CoA and sphingosine within SphK1 generates N-acetyl sphingosine (N-AS) that acetylates S565, of COX-2, leading to increased production of lipid mediators (LMs) that are subsequently converted to specialised pro-resolving mediators (SPMs); 3) N-AS generation is decreased in microglia in AD, leading to reduction of acetylated S565 of COX-2 and N-AS-triggered SPMs; 4) N-AS-triggered SPMs lead to resolution of neuroinflammation and up-regulation of reactive microglial genes linked to phagocytosis and reduction in AD pathology.

The data are novel and of interest for a wider field, in the situation when many clinical trials based

on targeting amyloidosis have failed. The work is generally convincing except for some parts:
1) the analysis of SPMs by ELISA's. First, these are not what is generally meant by ELISA's, i.e. sandwich ELISA's, but enzyme immune assays based on competition assays, where the SPM in the sample is expected to compete for the already bound SPM molecule. This type of assay is often used when it comes to small molecules which have poor antigenicity, such as the SPMs. The problem in the case of the SPMs is that they are very similar, and there is a high likelihood that the SPMs analysed in this paper with the competition assays cross-react and thus compete, not only in the respective assay, but also in the assays for the other SPMs. The authors should therefore test each SPM analysed in the competition assays for the other two SPMs and provide the data showing these analyses.

2) The authors claim in several instances in the Results as well as in the Discussion that they show that the N-AS-induced beneficial effects were mediated by SPMs, e.g. on p. 31 : "...the phagocytic ability of microglia was enhanced in N-AS-treated APP/PS1 mice via triggered SPMs. These results indicate that N-AS-triggered SPMs up regulate phagocytic abilities of microglia in AD mice, ..." and further down on p. 31: "...The N-AS-triggered SPMs derived from microglia unregulated several reactive microglial genes linked to phagocytosis, and down regulated several inflammatory and immune molecules in microglia of N-AS-treated APP/PS1 mice."

These and similar statements in the manuscript are not warranted of the the data shown, i.e. the evidence is circumstantial, and not directly showing that N-AS-induced SPMs mediate the effects seen after administration of N-AS. These statements need to be removed unless the authors will perform blocking studies to the effect that a direct link exist for SPMs mediating the responses to N-AS.

Considering the data presented, it is tempting to draw these conclusions, but the data presented do not give the evidence. Using circumstantial evidence will rather counteract the message and data of the paper, which would be a pity.

Reviewer #3 (Remarks to the Author):

Han et al. provides a comprehensive set of data on the protective role of N-acetyl sphingosine (N-AS) in AD mouse models. In a previous study, the group has reported the role of neuronal SphK1-acetyltransferase activity on COX2 and showed that it was reduced in AD neuron, exacerbating AD pathology (Nat Commun. 2018 Apr 16;9(1):1479.). Herein, the authors take a step further by finding that N-AS mediates the function of SphK1 in COX2 acetylation. They also pinpoint microglia as major cell type in the brain to generate N-AS. Intriguingly, treatment with N-AS increased acetylated COX2 and the production of SPMs in microglia from AD mice, leading to resolution of neuroinflammation, increased microglial phagocytosis, and mitigated behavioral deficits in two mouse models of AD. This study combines multiple approaches and provides convincing evidence in support of the beneficial role of enhancing microglial N-AS in AD-related pathologies.

I have minor questions and/or comments.

1. I would suggest including Suppl Fig 4a-b, Suppl Fig 5a and Suppl Fig 6a-6g as main Figures in the Fig 4 to indicate the roles of N-AS in AD model. Otherwise, Fig 4e-k should be moved to Suppl Fig 4.

2. For Fig 4d, authors should add the WB data for neuron to see whether the levels of ac-S565 in neuron are changed among WT, APP/PS1 and APP/PS1+N-AS groups.

3. For Suppl Fig 5k-n and Suppl Fig 7f-i, the immunofluorescence patterns of synaptophysin, MAP2, synapsin1 and PSD95 in APP/PS1 and 5xFAD mice need careful repeats as these proteins nearly lose their staining. As far as I know, so dramatic neuronal loss in 9-month-old APP/PS1 mice or 4-month-old 5xFAD mice has not been reported.

4. In Suppl Fig 4d, morphological characterization of microglia showed an activation-associated morphology after N-AS treatment in APP/PS1 mice. However, in Suppl Fig 5, the authors found N-AS decreased pro-inflammatory genes expression and increased anti-inflammatory genes expression. The authors need to explain the conflict between morphological changes and cytokines expression. Also, does N-AS specifically induce phagocytic morphology of the plaque-associated microglia? What happen to the non-plaque-associated microglia after N-AS treatment?

Reviewer #1 (Remarks to the Author)

The authors previously described in their recent published paper in Nature Comm 2018 the role of SphK1 in neurons in AD pathogenesis using animal models and human AD cells with a special focus on the impact on microglia mediated functions. They described the mechanism by which SphK1 leads to secretion of SRMs which involves the acetylation on serine 565 of COX2. The resulting increase of phagocytic microglia due to the release of proresolving lipids from neurons. In this former paper they reported that the enzyme SphK1 level and activity is decreased in AD APP/PS1 mice specifically in neurons but not in microglia or astrocytes, and that the neuronal enzyme activity, but not that in glial cells, affected the AD pathology and the cognitive deficit. Collectively, these results suggested and supported that increased neuronal SphK1 activity in APP/PS1 mice reduced Abeta load and improved cognitive deficits. They also reported that the increased neuronal SphK1 improve the neuroinflammation in AD brains. Overall this 2018 paper describes SphK1 dysfunction in neurons as the driving mechanism for microglia pathology and cognitive deficits in AD brain.

In this new paper, the authors reproduce faithfully the experimental design of the previous study of 2018 but focus their study on microglia. Moreover, they study more in-depth how SphK1 leads to acetylation of COX2 expanding the previous study of 2018 and including investigation on the involvement of N-acetyl sphingosine (N-AS). In this respect, the present study although interesting is incremental rather than proposing a new hypothesis for AD pathology.

The novel findings report that N-AS is reduced in microglia and neurons in AD, resulting in reduced SPMs release and the authors provide evidence for differentiating the molecular mechanism leading to reduced SPMs release and operating in microglia from the one operating in neurons. This was done in vitro by exposing microglia or neurons to Abeta.

The amelioration of AD pathology and behavioral outcomes were tested in AD mouse models treated with N-AS and they reproduced therapeutic effects similar to those previously reported in 2018 paper in mice with increased activation of SphK1 and COX2 acetylation in neurons. Findings in the present paper were additionally confirmed using a second AD mouse model (5xFAD).

Overall the study is certainly interesting but is rather incrementing the knowledge related to the original findings of the 2018 paper than establishing a new concept.

We would like to thank the reviewer for these comments concerning our manuscript. We agree with the reviewer's comment that the present paper increments the knowledge related to the original findings in our previous study (Nat Commun. 2018 Apr 16;9(1):1479.). We have also pointed out the novelty of the findings in this paper compared to our previous manuscript.

We previously demonstrated the role of neuronal SphK1-acetyltransferase activity on COX2 and showed that it was reduced in AD neurons, exacerbating AD pathology (Nat Commun. 2018 Apr 16;9(1):1479.). However, the mechanism of SphK1-mediated COX2 acetylation remained poorly understood, as well as its role in the pathogenesis of AD. In this study, we for the first time demonstrated the generation and roles of N-AS as an intermediate of SphK1-mediated COX2 acetylation. N-AS was first generated, followed by S565 acetylation of COX2 and SPM expression, in neurons and microglia of the CNS. This pathway was disrupted by different mechanisms in these two cell types upon exposure to A β . The reduction of N-AS generation in AD neurons was caused by decreased SphK1 activity, while reduced N-AS synthesis in AD microglia was caused by deficient acetyl-CoA, resulting in reduction of acetyl-S565 COX2 and SPM expression. Consistent with CNS cell studies, N-AS and S565 acetylation of COX2 decreased in neurons and microglia derived from APP/PS1 mice compared with WT mice. Interestingly, the levels of N-AS and acetyl-S565 COX2 were higher in microglia than in neurons, suggesting the generation and role of N-AS were more essential in microglia than neurons. These results indicated that N-AS might play important roles in microglia of AD, and that the reduced generation of N-AS might influence disease progression and/or pathogenesis in AD microglia. It was previously reported that the reduction of acetyl-CoA by A β led to mitochondrial dysfunction through oxidative damage in AD mice, resulting in defective microglia (Nature. 2006 Oct 19;443(7113):787-95.; Front Cell Neurosci. 2018 Jul 10;12:169.). However, the correlation of microglia dysfunction and acetyl-CoA metabolism has not been fully understood. In our study, we found that decreased N-AS generation was caused by deficient acetyl-CoA in AD microglia, followed by reduction of COX2 acetylation and SPM expression, resulting in neurodegenerative microglia. Based on previous studies and our new results, we propose that deficient acetyl-CoA metabolism by A β destroys microglia in AD through reduction of N-AS generation.

In addition, there is extensive literature supporting the importance of microglial regulation in the pathogenesis of AD (J Clin Invest. 2017 Sep 1;127(9):3240-3249.; J Clin Invest. 2015 Jan;125(1):350-64.; Nat Rev Immunol. 2018 Dec;18(12):759-772.), however the precise regulator(s) of microglia remains to be explored in AD. The SPMs, such as RvD1 and LXA₄, play an important role in the maintenance of phagocytic microglia (Pharmaceuticals (Basel).

2014 Nov 25;7(12):1028-48.), and are reduced in the AD brain, suggesting the possibility of SPMs as microglial regulators in AD. However, the mechanisms that underlie the regulation of microglia via SPMs have not been identified in most neuroinflammatory diseases, including AD. In a previous study, we reported the reduction of SphK1 activity and SPM secretion in AD neurons, which led to defective microglial phagocytosis and dysfunction of inflammation resolution (*Nat Commun.* 2018 Apr 16;9(1):1479.), indicating the indirect regulation of microglia by neurons. In this study, N-AS was mainly detected in microglia of N-AS-treated APP/PS1 mice rather than in neurons, and it directly regulated microglia via N-AS-triggered SPMs. The N-AS-triggered SPMs derived from microglia upregulated several reactive microglial genes related to phagocytosis, and downregulated several inflammatory and immune molecules in microglia of N-AS-treated APP/PS1 mice, leading to increase of phagocytic microglia. We also confirmed that N-AS-triggered SPMs derived from microglia resolve neuroinflammation via regulation of the immune system, similar to the function of SPMs reported in previous studies (*Immunity.* 2014 Mar 20;40(3):315-27.; *Nature.* 2014 Jun 5;510(7503):92-101.). Furthermore, N-AS-triggered SPMs, as well as N-AS generation and acetyl-S565 COX2, in microglia were higher than those in neurons, suggesting that the N-AS-triggered SPMs derived from microglia might play a more important role than those derived from neurons. Collectively, these results indicated that direct microglial regulation via N-AS-triggered SPMs derived from microglia might be more responsible for AD pathogenesis than indirect regulation of microglia through neurons, suggesting N-AS-triggered SPMs derived from microglia as direct regulators of pathology in AD. For more clarity, we also depicted the novel, advances of these findings and differences with the previous study in a schematic (**Additional Figure 1**).

(in *Nat Commun.* 2018 Apr 16;9(1):1479.)

(in *current study*)

Additional Figure 1. Different and advanced mechanisms in our previous study (*Nat Commun.* 2018 Apr 16;9(1):1479.) and the current study.

Importantly, we wish to point out that we proposed a therapeutic approach with practical possibilities for AD in this study rather than in our previous study. The findings in the previous study suggested a new therapeutic approach for AD may be possible by overexpressing SphK1 activity in neurons. However, because the body tightly controls the protein production levels, creating too many proteins or overexpressing a specific protein can be harmful to the cell in several ways, for example by activating or overloading specific biological pathways, disrupting regulation, or causing aggregation (eLife 2018;7:e39804.; Cell. 2009 Jul 10;138(1):198-208.; Cell. 2013 Jan 31;152(3):394-405.; Genome Res. 2013 Feb;23(2):300-11.). Moreover, to increase SphK1 specifically in AD neurons, in vivo gene transfer using AAV vectors may be considered one of the most efficient and promising protocols. Yet, the immune system remains a hurdle for in vivo gene transfer using AAV vectors (Mol Ther Methods Clin Dev. 2016 May 25;3:16034.). Unlike the therapeutic approach of overexpressing SphK1 in AD neurons, N-AS may be the therapeutic approach with the most practical potential for AD. In PK profiling of N-AS, a high concentration of N-AS was maintained in plasma and the brain, and N-AS had a high distribution in the brain. N-AS also had high binding affinity and induced high-level acetylation of the elevated COX2 in APP/PS1 microglia. Treatment with N-AS increased the levels of acetylated COX2 and N-AS-triggered SPMs in AD microglia, leading to the resolution of neuroinflammation, increase of microglial phagocytosis, and improvement of AD pathology. These results indicated that because N-AS has a high distribution in the brain and high binding affinity to the elevated COX2 in AD microglia, N-AS may well be delivered specifically to microglia in the AD brain, and this may lead to the restoration of AD pathology by direct regulation of microglia via N-AS-triggered SPM. Due to the aforementioned reasons, we considered that N-AS might be more useful for neuroinflammatory diseases including AD than the overexpression of SphK1 in AD neurons.

Overall, these results for the first time reveal a novel biosynthetic mechanism and function for N-AS, which lead to S565 acetylation of COX2 and the secretion of SPMs, and reveal the relation of N-AS with microglial regulation in AD pathogenesis. Furthermore, we suggest a novel potential therapy for neuroinflammatory diseases, such as AD, using N-AS that can be evaluated in the future rather than the proposal in the previous study. Thus, we think that the present paper provides incremental knowledge and establishes a new concept compared to our previous manuscript. We thank the reviewer for helping to make our findings more coherent and precise.

I consider the numerosity of samples in various experiments very low (n=3-4) for a sound statistical analysis of the data (see Figures 3-6). All or most of the vivo data in AD mouse models are in the Suppl information while the relevant ones showing the therapeutic effects of N-AS should appear in the main text.

We would like to thank the reviewer for these comments concerning our manuscript. We would like to respectfully point out that the sample sizes were determined by G-Power software (with $\alpha = 0.05$ and power of 0.8). In general, statistical methods were not used to re-calculate or predetermine the sample sizes. However, we agree with the reviewer's comment that the numerosity of the samples in various experiments was too low (n=3-4) for sound statistical analysis of the data, especially in Fig. 3-6. According to the reviewer's comment, we performed additional experiments to increase the numerosity of the sample size. Fig. 3-6, which had a low sample size (n=3-4), was replaced with the results obtained with a more reliable and proper sample size (n=6) by repeating the experiment. We confirmed that the results obtained by repeating the experiments were similar to the original data, and these data have been added to the revised manuscript. In addition, the RNA seq experiments shown in Fig. 4f-l in the revised manuscript were performed with sample pooling of two mice; therefore, 6-8 mice per group were used for RNA seq. We think the sample size used for RNA seq was adequate for statistical analysis, and this information has been added to the Figure legends of the revised manuscript.

As further suggested by the reviewer, we moved the in vivo data for the AD mouse models showing the therapeutic effects of N-AS (Supplementary Fig. 5-6; the original manuscript) to Fig. 6 of the revised manuscript, also reflecting the comments made by reviewer #3. We think that this relocation better supports the therapeutic effects of N-AS on the AD pathology including the reduction of amyloid plaque load and neuroinflammation and the restoration of cognitive function. We thank the reviewer for helping to make our findings more coherent and precise.

Reviewer #2 (Remarks to the Author)

This is an interesting paper, expressing the aim to further analyse the role of sphingosine kinase 1 (SphK1) in the acetyl-CoA-dependent acetylation of COX-2, and the mechanisms involved of this in Alzheimer's disease (AD). The main claims of the paper are that: 1) acetyl-CoA binds the ATP-binding site in SphK1; 2) reaction of acetyl-CoA and sphingosine within SphK1 generates N-acetyl sphingosine (N-AS) that acetylates S565, of COX-2, leading to increased production of lipid mediators (LMs) that are subsequently converted to specialised pro-resolving mediators (SPMs); 3) N-AS generation is decreased in microglia in AD, leading to reduction of acetylated S565 of COX-2 and N-AS-triggered SPMs; 4) N-AS-triggered SPMs lead to resolution of neuroinflammation and up-regulation of reactive microglial genes linked to phagocytosis and reduction in AD pathology.

The data are novel and of interest for a wider field, in the situation when many clinical trials based on targeting amyloidosis have failed. The work is generally convincing except for some parts.

We would like to thank the reviewer for these comments concerning our manuscript. We performed additional experiments and revised our manuscript according to the reviewer's suggestions (see below).

1) the analysis of SPMs by ELISA's. First, these are not what is generally meant by ELISA's, i.e. sandwich ELISA's, but enzyme immune assays based on competition assays, where the SPM in the sample is expected to compete for the already bound SPM molecule. This type of assay is often used when it comes to small molecules which have poor antigenicity, such as the SPMs. The problem in the case of the SPMs is that they are very similar, and there is a high likelihood that the SPMs analysed in this paper with the competition assays cross-react and thus compete, not only in the respective assay, but also in the assays for the other SPMs. The authors should therefore test each SPM analysed in the competition assays for the other two SPMs and provide the data showing these analyses.

We would like to thank the reviewer for these comments concerning our manuscript. In the original manuscript, we assessed the SPM secretion using sandwich ELISA (Fig. 3i and Supplementary Fig. 8d; in the original manuscript). However, we agree with the reviewer's comments that because SPMs are small and very similar molecules with poor antigenicity, each SPM should be analyzed using enzyme immune assays based on competition assays.

As suggested by the reviewer, we performed competitive ELISA of 15R-LXA₄, RvE1 and RvD1 in A β -affected neurons and microglia. We confirmed that the SPMs detected with the competition assays were decreased in A β -treated neurons and microglia, and improved after treatment with SphK1 and acetyl-CoA in A β -affected neurons and microglia, respectively, similar to restoration by treatment of N-AS. These data showed that A β led to the reduction of SPM secretion through the loss of SphK activity in neurons and the deficiency of acetyl-CoA in microglia, indicating different mechanisms for these two important cell types. Notably, the levels of SPM expression in microglia were higher than those in neurons, suggesting the N-AS-triggered SPMs are more essential in microglia than in neurons (Fig. 3i). In addition, we checked the SPM secretion using competition assays in A β -treated human microglia to assess the clinical importance of N-AS in microglia. SPM secretion was decreased in A β -treated human microglia compared with human microglia without treatment, and was restored by acetyl-CoA and N-AS treatment. This finding suggested that deficient acetyl-CoA induced by A β led to reduced secretion of N-AS-induced SPM, leading to microglial dysfunction in the AD environment (Supplementary Fig. 6d). These results also implied that the reduced secretion of N-AS-triggered SPMs might influence disease progression and/or pathogenesis in AD microglia, and may be restored with N-AS supplementation. To clarify our results, we replaced the sandwich ELISA results with competitive ELISA data in Fig. 3i and Supplementary Fig. 6d of the revised manuscript. We thank the reviewer for helping us

make our findings more coherent and precise.

Figure 3i in the revised manuscript. i, Quantification of 15R-LXA₄, RvE1, and RvD1 in microglia and neuron treated 10 μM Aβ or not in presence of N-AS, acetyl-CoA, sphingosine, or SphK1 each (n = 6 per group). i, One-way analysis of variance, Tukey's post hoc test. **P < 0.01, ***P < 0.001, ****P < 0.0001. All error bars indicate s.e.m.

Supplementary Figure 6d in the revised manuscript. d, Quantifications of 15R-LXA₄, RvE1, and RvD1 were performed using ELISA in the cell lysates of human microglia treated with or without 50 μ M A β in the presence or absence of N-AS, acetyl-CoA, or zileuton (zil) (n = 6 per group). d, and f, One-way analysis of variance, Tukey's post hoc test. *P < 0.05, **P < 0.01, ***P < 0.001, ****P < 0.0001. All error bars indicate s.e.m.

2) The authors claim in several instances in the Results as well as in the Discussion that they show that the N-AS-induced beneficial effects were mediated by SPMs, e.g. on p. 31 :
"..the phagocytic ability of microglia was enhanced in N-AS-treated APP/PS1 mice via triggered SPMs. These results indicate that N-AS-triggered SPMs up regulate phagocytic abilities of microglia in AD mice, ..."

and further down on p. 31: "...The N-AS-triggered SPMs derived from microglia unregulated several reactive microglial genes linked to phagocytosis, and down regulated several inflammatory and immune molecules in microglia of N-AS-treated APP/PS1 mice."

These and similar statements in the manuscript are not warranted of the the data shown, i.e. the evidence is circumstantial, and not directly showing that N-AS-induced SPMs mediate the effects seen after administration of N-AS. These statements need to be removed unless the authors will perform blocking studies to the effect that a direct link exist for SPMs mediating the responses to N-AS.

Considering the data presented, it is tempting to draw these conclusions, but the data presented do not give the evidence. Using circumstantial evidence will rather counteract the message and data of the paper, which would be a pity.

We thank the reviewer for this pertinent comment. As suggested by the reviewer, we have carefully revised our manuscript. In our study, we demonstrated that N-AS generation was mainly decreased in the microglia of APP/PS1 mice, and N-AS treatment in APP/PS1 mice led to the secretion of N-AS-triggered SPMs via S565 acetylation of COX2, resulting in beneficial effects through microglia regulation. We considered that N-AS-induced beneficial effects on microglia were mediated by N-AS-triggered SPMs. However, as mentioned by the reviewer, we did not show a direct link between these effects and N-AS-induced SPMs after the administration of N-AS.

According to the reviewer's comment, we checked whether N-AS-triggered SPMs directly mediate the effects seen after the administration of N-AS. Acetylated COX2 increases the production of 15-HETE, 18-HEPE, and 17-HDHA, which can be subsequently converted to SPMs, 15R-LXA₄, RvE1, and RvD1, respectively, in the presence of 5-lipoxygenase (5-LOX) (*Immunity*. 2014 Mar 20;40(3):315-27.; *Nature*. 2014 Jun 5;510(7503):92-101.). Based on these concepts, we considered that the inhibition of 5-LOX could block SPM production in the presence of N-AS. To block SPM production, we used the 5-LOX inhibitor, zileuton, which has been reported to reduce the production of SPMs by inhibiting 5-LOX (*J Pharmacol Exp Ther*. 2007 Jun;321(3):1154-60.; *Nat Rev Immunol*. 2015 Aug;15(8):511-23.; *Immunity*. 2018 May 15;48(5):1006-1013.) in the presence of N-AS. When zileuton was incubated with N-AS in A β -treated human microglia, no significant differences in N-AS-induced COX2

acetylation were observed compared with the effects in N-AS and A β -treated microglia without zileuton (Supplementary Fig. 6b and c). However, SPM secretion was significantly reduced with the treatment of zileuton, indicating that zileuton specifically inhibited SPM secretion, rather than affecting COX2 acetylation by N-AS (Supplementary Fig. 6d). Interestingly, the inhibition of SPM secretion by zileuton in the presence of N-AS blocked the restoration of phagocytic capacity in A β -treated microglia via N-AS, indicating that N-AS-triggered SPMs directly mediate the positive effects in microglia (Supplementary Fig. 6e and f). These results suggested N-AS-triggered SPMs were responsible for direct regulation of microglia including phagocytosis in a human AD environment, supporting the potential clinical effects in AD patients. To clarify our results, we have added these data in Supplementary Fig. 6b-f and Supplementary Videos 1-6 of the revised manuscript.

We would like to thank the reviewer for all their comments concerning our manuscript.

Supplementary Figure 6b-f in the revised manuscript. **b**, Western blot analysis for ac-S565 and total COX2 in human microglia treated 50 μM Aβ or not in presence or absence of N-AS or acetyl-CoA or zileuton (zil) (n = 6 per group). **c**, Colocalization of microglia (Iba1, red) with ac-S565 (green) and COX2 (blue) and quantification. (n = 6 per group, Scale bars, 50 μm). **d**, Quantifications of 15R-LXA₄, RvE1, and RvD1 were detected by ELISA in cell lysate of human microglia treated 50 μM Aβ or not in presence or absence of N-AS or acetyl-CoA or zileuton (n = 6 per group). **e**, Representative images from live-cell imaging at various times after the administration of beads. Scale bars, 30 μm. **f**, Analysis of beads uptake. The amount of time taken to phagocytose the first beads (top), the time between phagocytosing the first and second beads (middle) and the final number of beads phagocytosed in 120 min (bottom) (n = 6 per group). **b-d**, and **f**, One-way analysis of variance, Tukey's post hoc test. **P < 0.01, ***P < 0.001, ****P < 0.0001. All error bars indicate s.e.m.

Reviewer #3 (Remarks to the Author):

Han et al. provides a comprehensive set of data on the protective role of N-acetyl sphingosine (N-AS) in AD mouse models. In a previous study, the group has reported the role of neuronal SphK1-acetyltransferase activity on COX2 and showed that it was reduced in AD neuron, exacerbating AD pathology (Nat Commun. 2018 Apr 16;9(1):1479.). Herein, the authors take a step further by finding that N-AS mediates the function of SphK1 in COX2 acetylation. They also pinpoint microglia as major cell type in the brain to generate N-AS. Intriguingly, treatment with N-AS increased acetylated COX2 and the production of SPMs in microglia from AD mice, leading to resolution of neuroinflammation, increased microglial phagocytosis, and mitigated behavioral deficits in two mouse models of AD. This study combines multiple approaches and provides convincing evidence in support of the beneficial role of enhancing microglial N-AS in AD-related pathologies.

We thank the reviewer for these positive comments concerning our manuscript.

I have minor questions and/or comments.

1. I would suggest including Suppl Fig 4a-b, Suppl Fig 5a and Suppl Fig 6a-6g as main Figures in the Fig 4 to indicate the roles of N-AS in AD model. Otherwise, Fig 4e-k should be moved to Suppl Fig 4.

We would like to thank the reviewer for these comments concerning our manuscript, which are similar to those made by reviewer #1. According to the reviewer's comment, we have moved **Supplementary Fig. 4a-b, 5a and 6a-g** in the original manuscript to **Fig. 5a-b and 6** in the revised manuscript to better support the therapeutic effects of N-AS on the AD pathology. We also transferred **Fig. 4e-k** in the original manuscript to **Supplementary Fig. 4b-f** in the revised manuscript.

We wish to point out that many studies have reported that the restoration of microglia increases A β phagocytosis in AD, leading to the amelioration of AD pathology, neuropathology, and behavior (*J Clin Invest.* 2017 Sep 1;127(9):3240-3249.; *Nat Commun.* 2018 Apr 16;9(1):1479.). We also confirmed that N-AS-triggered SPMs positively regulate microglia phagocytic functions and the immune response, resulting in the reduction of A β deposits and the resolution of inflammation in the AD environment. We want to focus on the improvement of AD pathology by positive microglia regulation via N-AS-triggered SPMs. For this reason, we first described the role of N-AS-triggered SPM, i.e. regulating microglia phagocytic functions and immune response, prior to the restoration of the AD pathology in APP/PS1 mice treated with N-AS. Therefore, we think that restoration of the AD pathology in N-AS-injected AD models is more appropriately presented in Fig. 6 than in Fig. 4, and hope the reviewer understands our preference.

2. For Fig 4d, authors should add the WB data for neuron to see whether the levels of ac-S565 in neuron are changed among WT, APP/PS1 and APP/PS1+N-AS groups.

We would like to thank the reviewer for these comments concerning our manuscript. According to the reviewer's comment, we performed western blotting to determine the levels of ac-S565 COX2 in neurons derived from WT, APP/PS1, and APP/PS1 mice treated with N-AS. The expression of ac-S565 COX2 was decreased in APP/PS1 neurons compared with WT neurons, and was restored in APP/PS1 mice treated with N-AS neurons, similar to the results for ac-S565 COX2 in microglia. However, the levels of N-AS-mediated COX2 acetylation were better recovered in the microglia of APP/PS1 mice compared with those in

the neurons in APP/PS1 mice treated with N-AS (Fig. 4d). These data indicated that microglia were the likely target cells responsible for AD pathogenesis via N-AS. To clarify our results, we added these data in Fig. 4d of the revised manuscript.

Figure 4d in the revised manuscript. d, Western blot analysis for ac-S565 and total COX2 in microglia and neuron derived from WT, APP/PS1, and APP/PS1 injected with N-AS mice (n = 6 per group). All data analysis was done at 9-mo-old mice. d, One-way analysis of variance. *P < 0.05, ***P < 0.001. All error bars indicate s.e.m.

3. For Suppl Fig 5k-n and Suppl Fig 7f-i, the immunofluorescence patterns of synaptophysin, MAP2, synapsin1 and PSD95 in APP/PS1 and 5xFAD mice need careful repeats as these proteins nearly lose their staining. As far as I know, so dramatic neuronal loss in 9-month-old APP/PS1 mice or 4-month-old 5xFAD mice has not been reported.

We would like to thank the reviewer for these comments. According to the reviewer's comment, we have carefully reviewed and reconfirmed our immunofluorescence images and data about synaptic markers (Supplementary Fig. 5k-n and 7f-i in original manuscript).

However, we obtained similar results with the immunofluorescence patterns in the original manuscripts, showing dramatic reduction of the synaptic marker in APP/PS1 and 5xFAD mice.

Therefore, to better assess the reliability of our results, we reviewed the literature and found that many studies reported decreased levels for synaptic markers including synaptophysin, MAP2, synapsin1, and PSD95 in 9-month-old APP/PS1 mice and 4-month-old 5xFAD mice, determined using western blotting for the synaptic markers rather than immunofluorescence (*Neural Regen Res.* 2016 Oct;11(10):1617-1624; *Nat Neurosci.* 2013 Sep;16(9):1299-305; *PLoS One.* 2015 Aug 18;10(8):e0135686; *J Neurochem.* 2015 Apr;133(1):38-52). Based on these studies, we checked the synaptic markers using western blotting for synaptophysin, MAP2, synapsin 1, and PSD95 in 9-month-old APP/PS1 mice and 4-month-old 5xFAD mice. We confirmed that the expression of synaptic markers was decreased in 9-month-old APP/PS1 mice and 4-month-old 5xFAD mice compared with age-matched control mice, and was restored with N-AS treatment (**Supplementary Fig. 7a-d and 8f-l**). The data obtained with western blotting for synaptic proteins showed a decrease of about 50% in 9-month-old APP/PS1 mice and 4-month-old 5xFAD mice compared with age-matched control mice, unlike immunofluorescence-based analysis showing dramatic synaptic protein loss. We think the reduction of synaptic markers may be observed in 9-month-old APP/PS1 mice and 4-month-old 5xFAD mice, and the results obtained from western blotting rather than immunofluorescence were more appropriate to make our findings more coherent and precise. Due to the aforementioned reasons, we have replaced the data for immunofluorescence (**Supplementary Fig. 5k-n and 7f-i** in original manuscript) with the results of western blotting in **Supplementary Fig. 7a-d and 8f-i** in the revised manuscript.

Supplementary Figure 7a-d in the revised manuscript. a-d, Western blot analysis for synaptophysin (a; n = 6 per group), MAP2 (b; n = 6 per group), synapsin1 (c; n = 6 per group) and PSD95 (d; n = 6 per group) in cortex of WT, APP/PS1, and APP/PS1 mice treated with N-AS. All data analysis was done at 9-mo-old mice. a-d, One-way analysis of variance, Tukey's post hoc test. **P < 0.01, ***P < 0.001, ****P < 0.0001. All error bars indicate s.e.m.

Supplementary Figure 8f-i in the revised manuscript. f-i, Western blot analysis for synaptophysin (f; n = 6 per group), MAP2 (g; n = 6 per group), synapsin1 (h; n = 6 per group), and PSD95 (i; n = 6 per group) in cortex of WT, 5xFAD, and 5xFAD treated with N-AS. All data analysis was done at 4-mo-old mice. b and c, Student's t test. d-i, One-way analysis of variance, Tukey's post hoc test. **P < 0.01, ***P < 0.001. All error bars indicate s.e.m.

4. In Suppl Fig 4d, morphological characterization of microglia showed an activation-associated morphology after N-AS treatment in APP/PS1 mice. However, in Suppl Fig 5, the authors found N-AS decreased pro-inflammatory genes expression and increased anti-inflammatory genes expression. The authors need to explain the conflict between morphological changes and cytokines expression. Also, does N-AS specifically induce phagocytic morphology of the plaque-associated microglia? What happen to the non-plaque-associated microglia after N-AS treatment?

We would like to thank the reviewer for these comments concerning our manuscript. We would also like to respectfully point out that a recent study reported that activated microglia have considerably variable morphology and gene expression (Front Mol Neurosci. 2017; 10: 191.). Disturbances of brain homeostasis can induce rapid and profound changes in microglial morphology, gene expression, and gene function. These events define the so-called "microglial activation" and include changes in gene expression, reorganization of surface molecules for interactions with the extracellular environment and neighboring cells, and the release of soluble factors acting as pro- or anti-inflammatory factors. Microglia can also become phagocytic to remove tissue debris and damaged cells. The different functional phases of microglia activation are set out on a morphological, molecular, and functional basis. The microglial activation is thus a highly regulated process (Neurol Res. 2005 Oct;27(7):685-91.; Nat Neurosci. 2007 Nov;10(11):1387-94.; CNS Neurol Disord Drug Targets. 2010 Apr;9(2):174-91.; Acta Neuropathol. 2010 Jan;119(1):89-105.).

In neurodegenerative diseases, including AD, microglia are known to have two major types of activated morphology, namely neurodegenerative phenotype microglia and phagocytic microglia. The number of neurodegenerative phenotype microglia, which release pro-inflammatory cytokines, is increased, while that of phagocytic microglia, which remove debris including A β and release anti-inflammatory cytokines, is reduced (Nat Med. 2017 Sep 8;23(9):1018-1027). In our study, when we performed morphological characterization of the microglia surrounding A β , we found that the neurodegenerative phenotype was reduced, and the phagocytic morphology (amoeboid microglial morphology) was more evident in APP/PS1 mice treated with N-AS than in APP/PS1 mice treated with vehicle (Supplementary Fig. 5b). These results indicated that when N-AS is administered in APP/PS1 mice, the neurodegenerative phenotype microglia, which release pro-inflammatory cytokines, were reduced while the phagocytic microglia, which release anti-inflammatory cytokines, were elevated. These changes in the activated microglial morphology with N-AS treatment may cause decreased pro-inflammatory gene expression and increased anti-inflammatory gene expression in APP/PS1 mice (Fig. 6j). Taken together, our results suggested that

neuroinflammation was reduced with N-AS treatment in APP/PS1 mice, converting activated microglial morphology from neurodegenerative microglia to phagocytic microglia.

As further suggested by the reviewer, we performed additional experiments to check the non-plaque-associated microglia in APP/PS1 mice and N-AS-treated APP/PS1 mice. We confirmed that in APP/PS1 mice and N-AS-treated APP/PS1 mice, the morphology of the non-plaque-associated microglia presented a ramified morphology, which is characterized by a small cell body (*Psychopharmacology (Berl)*. 2016 May;233(9):1543-57.). This morphology did not differ between the groups (**Additional Fig. 2**). A previous study reported that the expression of AD-related markers was similarly low in the non-plaque associated cells of AD brains. Only the A β plaque-associated microglia are hyperreactive in their immune response and phagocytosis in the transgenic AD mice (*Neurobiol Aging*. 2017 Jul;55:115-122.). Based on the previous study and our results, we concluded that the plaque-associated microglia are hyper-reactivated in APP/PS1 mice, and N-AS treatment in APP/PS1 mice changed the microglial morphology from neuroinflammatory to phagocytic only in the plaque-associated microglia, not in the non-plaque-associated microglia.

We thank the reviewer for helping to make our findings more coherent and precise.

Additional Figure 2. Morphology of non-plaque-associated microglia in cortex of APP/PS1 and APP/PS1 mice injected with N-AS. Imaris-based three-dimensional images (Scale bars, 10 μm) of microglia surrounding A β . Bottom, Imaris-based automated quantification of microglial morphology (n = 7-8 mice per group).

Reviewers' comments:

Reviewer #2 (Remarks to the Author):

The authors are thanked for their responses to the comments by this reviewer.
Concerning the 1st comment, i.e. regarding ELISA's and EIA's for SPMs:

The authors have performed both sandwich ELISA's and competitive EIA's for the SPMs with similar results. It is good that the results are similar with the different techniques. However, the authors have not performed what was asked for, i.e. to test whether

- a) the ELISA and/or EIA for LXA4 will detect RvD1 or RvE1,
- b) the ELISA and/or EIA for RvD1 will detect LXA4 or RvE1,
- c) the ELISA and/or EIA for RvE1 will detect LXA4 or RvD1.

This should be done with different concentrations of e.g. RvE1 in the RvD1 assay, e.g. using the standard curve for the SPM tested. In this way it would be seen whether the RvD1 assay also detects RvE1.

The question raised regarding cross-reactivity concerns both ELISA and EIA, since the SPMs are small molecules and with similarities and therefore cross-reactivity may be a concern, and in addition, there could also be cross-reactivity with other lipids (see comment below).

It is preferable to ascertain the data by LC-LC-MS analysis.

Regarding comment 2:

The authors have performed new experiments with an inhibitor of 5-LOX with interesting findings that when adding the inhibitor the "restoration of phagocytic capacity in A β -treated microglia" was blocked, at the same time as the levels of SPMs were reduced. However, the inhibitor also blocks synthesis of leukotrienes.

It would be good if the authors could discuss the possibility that blocking 5-LOX may result in the reduction of other mediators that may have effects on phagocytosis. This is especially since the measurements using ELISA or EIA could well be detecting other lipids than those intended for the assays. The producers of the assays do state that they cannot guarantee that the assays are completely specific. Thus, this could also be mentioned by the authors.

Reviewer #3 (Remarks to the Author):

The authors have adequately addressed most of the comments that have been raised in a previous round of review. However, there's poor evidence supporting their claim that N-AS-triggered SPMs derived from microglia as direct regulators of pathology in AD. In Fig. 4d, the levels of N-AS-mediated COX2 acetylation both recovered ~ 2 fold in the microglia and neurons of APP/PS1 mice treated with N-AS. PLX3397 or 5622-depletion of microglia is a helpful approach to address whether the N-AS-SPMs-effects are more dependent on microglia.

Reviewer #2 (Remarks to the Author):

The authors are thanked for their responses to the comments by this reviewer.

Concerning the 1st comment, i.e. regarding ELISA's and EIA's for SPMs:

The authors have performed both sandwich ELISA's and competitive EIA's for the SPMs with similar results. It is good that the results are similar with the different techniques.

However, the authors have not performed what was asked for, i.e. to test whether

a) the ELISA and/or EIA for LXA4 will detect RvD1 or RvE1,

b) the ELISA and/or EIA for RvD1 will detect LXA4 or RvE1,

c) the ELISA and/or EIA for RvE1 will detect LXA4 or RvD1.

This should be done with different concentrations of e.g. RvE1 in the RvD1 assay, e.g. using the standard curve for the SPM tested. In this way it would be seen whether the RvD1 assay also detects RvE1.

The question raised regarding cross-reactivity concerns both ELISA and EIA, since the SPMs are small molecules and with similarities and therefore cross-reactivity may be a concern, and in addition, there could also be cross-reactivity with other lipids (see comment below).

It is preferable to ascertain the data by LC-LC-MS analysis.

We would like to thank the reviewer for these comments concerning our manuscript. We agree with the reviewer's comment that sandwich ELISA and competitive EIA raised regarding cross-reactivity concerns. As suggested by the reviewer, LC-MS/MS allow even to distinguish between very structurally similar biomolecules when for the identification and quantitation of biomolecules, and many analytes can be analyzed at a time for a single LC-MS/MS analysis of a sample. Therefore, to resolve the question regarding cross-reactivity concerns of SPMs, SPMs should be analyzed using LC-MS/MS analysis.

Accordingly, we performed LC-MS/MS analysis of 15R-LXA₄, RvE1 and RvD1 in mouse neuron, mouse microglia and human microglia (Additional Fig. 1). The SPMs detected by LC-MS/MS were decreased in A β -treated mouse neurons and microglia, and improved after treatment with SphK1 and acetyl-CoA in A β -affected mouse neurons and microglia respectively, similar to restoration by treatment of N-AS. These data showed that A β led to reduction of SPM secretion through the loss of SphK activity in neurons and the deficiency of acetyl-CoA in microglia, indicating different mechanisms for these two important cell types.

Notably, the levels of SPMs expression in microglia were higher than those in neurons, suggesting the N-AS-triggered SPM was more essential in microglia than neurons (Fig. 3i). In addition, we have checked SPMs secretion using LC-MS/MS in A β -treated human microglia to assess the clinical importance of N-AS in microglia. SPM secretion was decreased in A β -treated human microglia compared with human microglia without treatment, and restored by acetyl-CoA and N-AS treatment, suggesting that deficient acetyl-CoA induced by A β led to reduced secretion of N-AS-induced SPM, leading to microglial dysfunction in the AD environment (Supplementary Fig. 6d). These results proposed that the reduced secretion of N-AS-triggered SPMs might influence disease progression and/or pathogenesis in AD microglia, and might be treated by N-AS supplementation.

In LC-MS/MS analysis, we obtained similar results SPMs with sandwich ELISA's and competitive EIA's for the SPMs, and think that LC-MS/MS analysis solved cross-reactivity concerns of SPMs. To clarify our results, we have replaced competitive ELISA data results to LC-MS/MS data in Fig. 3i and Supplementary Fig. 6d of the revised manuscript. We thank the reviewer for helping us make our findings more coherent and precise.

Additional Figure 1. Representative chromatograms of blank, SPMs standard, and SPMs in cell lysate of mouse neuron, mouse microglia, and human microglia. 15R-LXA₄ (left), RvE1 (middle), and RvD1 (right).

Figure 3i in the revised manuscript. i, Quantification of 15R-LXA₄, RvE1, and RvD1 in microglia and neuron treated 10 μM Aβ or not in presence of N-AS, acetyl-CoA, sphingosine, or SphK1 each (n = 6 per group). i, One-way analysis of variance, Tukey's post hoc test. ****P <0.0001. All error bars indicate s.e.m.

Supplementary Figure 6d in the revised manuscript. d, Quantifications of 15R-LXA₄, RvE1, and RvD1 were detected by LC-MS/MS in cell lysate of human microglia treated 50 μM Aβ or not in presence or absence of N-AS or acetyl-CoA or zileuton (zil) (n = 6 per group). d, and f, One-way analysis of variance, Tukey's post hoc test. *P < 0.05, **P < 0.01, ***P < 0.001, ****P < 0.0001. All error bars indicate s.e.m.

Regarding comment 2:

The authors have performed new experiments with an inhibitor of 5-LOX with interesting findings that when adding the inhibitor the "restoration of phagocytic capacity in A β -treated microglia" was blocked, at the same time as the levels of SPMs were reduced. However, the inhibitor also blocks synthesis of leukotrienes.

It would be good if the authors could discuss the possibility that blocking 5-LOX may result in the reduction of other mediators that may have effects on phagocytosis. This is especially since the measurements using ELISA or EIA could well be detecting other lipids than those intended for the assays. The producers of the assays do state that they cannot guarantee that the assays are completely specific. Thus, this could also be mentioned by the authors.

We would like to thank the reviewer for these comments concerning our manuscript. To check whether N-AS-triggered SPMs directly mediate the effects seen after administration of N-AS, we undertook 5-LOX inhibitor, zileuton, which was reported that it reduced the production of SPMs by inhibiting 5-LOX (J Pharmacol Exp Ther. 2007 Jun;321(3):1154-60.; Nat Rev Immunol. 2015 Aug;15(8):511-23.; Immunity. 2018 May 15;48(5):1006-1013.), in the presence of N-AS. We confirmed that inhibition of SPM secretion by zileuton blocked restoration of phagocytic capacity in A β -treated microglia via N-AS, indicating that N-AS-triggered SPMs directly mediate the positive effects in microglia.

However, as mentioned by the reviewer, the zileuton also reduce formation of leukotriene (e.g. LTB₄) and inhibit prostaglandin (e.g. PGE₂) production by interfering at the level of arachidonic acid release (Br J Pharmacol. 2010 Oct;161(3):555-70.). Therefore, we cannot completely rule out the possibility that blocking 5-LOX may result in the reduction of other mediators that may have effects on phagocytosis. To investigate that the only N-AS-triggered SPMs, not the other lipid mediators, directly mediate positive regulation of microglial phagocytosis, we performed lipid mediator-SPM profiles in A β -treated microglia with N-AS in the presence of zileuton. When zileuton was incubated with N-AS in A β -treated human microglia, we confirmed reduction of SPMs and PGE₂, although LTB₄ was not detected (Supplementary Table 3). Also, zileuton treatment in presence of N-AS blocked restoration of phagocytic capacity in A β -treated microglia (Supplementary Fig. 6e, f). Both SPMs and PGE₂, which decreased by zileuton, play an important role in phagocytic ability of microglia, and phagocytosis is restored through the increase of SPMs secretion and inhibition of PGE₂ synthesis (Front Cell Infect Microbiol. 2013; 3: 45.; J Clin Invest. 2015 Jan;125(1):350-64.; Nat Commun. 2018 Apr 16;9(1):1479.). In our results, although the levels of PGE₂ were decreased, the phagocytic ability was not restored in N-AS and A β -treated microglia in the presence of zileuton. These results supported that blocked

restoration of phagocytic capacity in A β -treated microglia was caused by inhibition of N-AS-triggered SPMs via zileuton treatment, suggesting that the N-AS-triggered SPMs, rather than the other lipid mediators, were more responsible direct regulation of microglia including phagocytosis in a human AD environment. To clarify our results, we have added these data in **Supplementary Table 3** of the revised manuscript, and discussed other lipid mediators decreased by 5-LOX inhibition that may have effects on phagocytosis in the **Discussion section** of the revised manuscript.

We would like to thank reviewer for all their comments concerning our manuscript.

LM in human microglia (ng/ml*10⁶cell)

Bioactive metabolome	Q1	Q3	LM in human microglia (ng/ml*10 ⁶ cell)						
			Control	Control + N-AS	Control + Zil + N-AS	A β	A β + Acetyl-CoA	A β + N-AS	A β + Zil + N-AS
AA bioactive metabolome									
PGH ₂ /PGE ₂	351	271	116.5±1.1	105.8±1.1	79.5±0.9*****	78.7±0.8**	103.1±1.0 [#]	95.7±0.9 [#]	76.0±1.0** [€]
TxB ₂	369	169	ND	ND	ND	ND	ND	ND	ND
11-HETE	319	167	168.5±1.8	161.8±0.9	162.5±0.2	174.9±2.0	151.2±0.5	148.4±1.9	144.4±0.9
LTB ₄	335	195	ND	ND	ND	ND	ND	ND	ND
15-HETE	319	219	647.1±7.0	625.5±6.8	610.2±4.4	564.3±4.4**	609.4±1.9 [#]	625.4±1.7 [#]	612.5±2.7 [#]
15R-LXA ₄	351	115	31.8±0.4	28.7±0.3	0.03±0.0*****	0.03±0.0**	29.6±0.3 [#]	19.8±0.2 [#]	0.02±0.0*** ^{€€€}
EPA bioactive metabolome									
PGE ₃ /PGD ₃	349	269	142.9±2.2	113.1±1.6	130.0±1.9	163.9±2.5	158.7±2.5	132.9±2.0	146.7±0.6
TxB ₃	367	169	ND	ND	ND	ND	ND	ND	ND
11-HEPE	317	167	484.4±5.8	451.8±4.6	483.1±2.4	490.2±2.4	467.8±4.5	371.7±4.0	480.6±2.3
15-HEPE	317	219	298.9±5.0	284.1±4.0	273.7±4.2	272.1±7.3	297.5±3.8	237.9±3.7	268.2±2.1
18-HEPE	317	215	1729.2±24.1	1636.9±19.7	1650.0±7.6	1515.3±10.4**	1881.1±22.9 [#]	1763.6±30.4 [#]	1668.4±4.3 [#]
RvE1	349	195	22.3±0.3	18.5±0.4	0.01±0.0*****	0.01±0.0***	14.5±0.2 [#]	8.6±0.3 [#]	0.01±0.0*** ^{€€€}
DHA bioactive metabolome									
13-HDHA	343	193	146.6±2.1	133.9±2.2	137.6±1.6	128.2±1.8	147.4±1.0	133.0±1.8	140.8±2.9
17-HDHA	343	281	693.3±8.5	633.8±9.2	642.3±12.4	561.8±2.7**	720.8±5.3 [#]	707.6±8.2 [#]	705.4±6.0 [#]
RvD1	375	141	2.5±0.02	2.36±0.02	0.01±0.0*****	0.01±0.0***	2.75±0.04 [#]	2.13±0.02 [#]	0.01±0.0*** ^{€€€}
RvD2	375	277	ND	ND	ND	ND	ND	ND	ND
RvD3	375	147	ND	ND	ND	ND	ND	ND	ND

Human microglia were treated with N-AS, acetyl-CoA or Zil in presence of A β 50 μ M or not (n = 6 per group). All data are expressed as mean±s.e.m. One-way analysis of variance, Tukey's post hoc test. ***P* < 0.01, ****P* < 0.001 versus control. [#]*P* < 0.01, [#][#]*P* < 0.001 versus A β . ^{*}*P* < 0.05, ^{**}*P* < 0.01, ^{***}*P* < 0.001 versus control+N-AS. [€]*P* < 0.05, ^{€€€}*P* < 0.001 versus A β +N-AS.

Supplementary Table 3 in the revised manuscript. Human Microglia LM-SPM profiles from Control treated with N-AS, acetyl-CoA or Zil in presence of A β 50 μ M or not.

Reviewers' comments:

Reviewer #2 (Remarks to the Author):

The topic of this manuscript is very interesting and most of the data warrant publication. Therefore it is a pity that the authors have not performed the minor check of cross-reactivity between the ELISA's (or EIA's) for the different lipid mediators as delineated below and asked for by this reviewer, i.e. to test if

- a) the ELISA and/or EIA for LXA4 will detect RvD1 or RvE1,
- b) the ELISA and/or EIA for RvD1 will detect LXA4 or RvE1,
- c) the ELISA and/or EIA for RvE1 will detect LXA4 or RvD1.

 This should be done, and a simple way is to add different concentrations of e.g. RvE1 in the RvD1 assay, e.g. using the standard curve for the SPM tested. In this way it would be seen whether the RvD1 assay also detects RvE1.

These tests are easy to perform and do not require experimental samples.

The question raised regarding cross-reactivity concerns both ELISA and EIA, since the SPMs are small molecules and with similarities and therefore cross-reactivity may be a concern, and in addition, there could also be cross-reactivity with other lipids (see comment below).

It is preferable, but not mandatory to ascertain the data by LC-MS-MS analysis.

Reviewer #2 (Remarks to the Author):

The topic of this manuscript is very interesting and most of the data warrant publication. Therefore it is a pity that the authors have not performed the minor check of cross-reactivity between the ELISA's (or EIA's) for the different lipid mediators as delineated below and asked for by this reviewer, i.e. to test if

a) the ELISA and/or EIA for LXA4 will detect RvD1 or RvE1,

b) the ELISA and/or EIA for RvD1 will detect LXA4 or RvE1,

c) the ELISA and/or EIA for RvE1 will detect LXA4 or RvD1.

This should be done, and a simple way is to add different concentrations of e.g. RvE1 in the RvD1 assay, e.g. using the standard curve for the SPM tested. In this way it would be seen whether the RvD1 assay also detects RvE1.

These tests are easy to perform and do not require experimental samples.

The question raised regarding cross-reactivity concerns both ELISA and EIA, since the SPMs are small molecules and with similarities and therefore cross-reactivity may be a concern, and in addition, there could also be cross-reactivity with other lipids (see comment below).

It is preferable, but not mandatory to ascertain the data by LC-MS-MS analysis.

We would like to thank the reviewer for these comments concerning our manuscript. According to the reviewer's comment, we performed check of cross-reactivity using competitive EIA of 15R-LXA₄, RvE1, and RvD1. The RvE1 and RvD1 were added with various concentrations in 15R-LXA₄ assay to examine cross-reactivity for the different lipid mediators in 15R-LXA₄ assay. We confirmed that the OD value of 15R-LXA₄ was decreased in a concentration dependent fashion, but not RvE1 and RvD1. This result indicates that competitive EIA of 15R-LXA₄ did not detect the different lipid mediators including RvE1 and RvD1 (Additional Fig. 1a). We also assessed cross-reactivity in RvE1 and RvD1 assay, similar to 15R-LXA₄ assay, applying 15R-LXA₄ and RvD1 in RvE1 assay, and 15R-LXA₄ and RvE1 in RvD1 assay. In RvE1 assay, the OD value of RvE1 was reduced with increasing concentrations of RvE1, while OD value of 15R-LXA₄ and RvD1 was not changed according to concentrations of RvE1 (Additional Fig. 1b). Moreover, the OD value for RvD1 decreased as the concentration of RvD1 increased in RvD1 assay, whereas the OD value for 15R-LXA₄ and RvE1 did not altered (Additional Fig. 1c). Similar to 15R-LXA₄, the competitive EIA of RvE1 and RvD1 also showed that they did not detect the different lipid mediators. Collectively, these results suggest that cross-reactivity for the different lipid mediators was not presented in competitive EIA of 15R-LXA₄, RvE1, and RvD1.

Additional Figure 1. Detection of cross-reactivity in competitive EIA of 15R-LXA₄, RvE1 and RvD1. (a) 15R-LXA₄, (b) RvE1 and (c) RvD1.

To more clearly resolve the question regarding cross-reactivity concerns of SPMs, we analyzed SPMs using LC-MS/MS. SPMs are defined by unique and characteristic retention time in LC-MS/MS analysis, indicating that LC-MS/MS analysis did not show cross-reactivity for the different lipid mediators (Additional Fig. 2a). Accordingly, we performed LC-MS/MS analysis of 15R-LXA₄, RvE1 and RvD1 in mouse neuron, mouse microglia, and human microglia (Additional Fig. 2b).

As we already showed in the previous 2nd revision, the SPMs detected by LC-MS/MS were decreased in A β -treated mouse neurons and microglia, and improved after treatment with SphK1 and acetyl-CoA in A β -affected mouse neurons and microglia respectively, similar to restoration by treatment of N-AS. These data showed that A β led to reduction of SPM secretion through the loss of SphK activity in neurons and the deficiency of acetyl-CoA in microglia, indicating different mechanisms for these two important cell types. Notably, the levels of SPMs expression in microglia were higher than those in neurons, suggesting the N-AS-triggered SPM was more essential in microglia than neurons (Fig. 3i). In addition, we have checked SPMs secretion using LC-MS/MS in A β -treated human microglia to assess the clinical importance of N-AS in microglia. SPM secretion was decreased in A β -treated human microglia compared with human microglia without treatment, and restored by acetyl-CoA and N-AS treatment, suggesting that deficient acetyl-CoA induced by A β led to reduced secretion of N-AS-induced SPM, leading to microglial dysfunction in the AD environment (Supplementary Fig. 6d). These results proposed that the reduced secretion of N-AS-triggered SPMs might influence disease progression and/or pathogenesis in AD microglia, and might be treated by N-AS supplementation.

We thank the reviewer for helping us make our findings more coherent and precise.

Additional Figure 2. **a**, Retention time of 15R-LXA₄, RvE1 and RvD1. **b**, Representative chromatograms of blank, SPMs standard, and SPMs in cell lysate of mouse neuron, mouse microglia and human microglia. 15R-LXA₄ (left), RvE1 (middle) and RvD1 (right).

REVIEWERS' COMMENTS:

Reviewer #2 (Remarks to the Author):

It is acknowledged that the authors have now performed the cross-reactivity test for the analyses of the lipid mediators. It has to be mentioned that this test indeed revealed cross-reactivity for RvD1 in the LXA4 ELISA, as well as for LXA4 in the RvD1 ELISA. These cross-reactivities are not minor, and the interpretation made in the answers by the authors is therefore not correct. However, it appears that all of the analyses of lipid mediators are now verified by LC-MS-MS analyses, and this reviewer does not object to acceptance of the manuscript on the basis of the lipid mediator analysis.

If, on the contrary, some analysis of lipid mediators is still performed by ELISA (or EIA), the authors must discuss the issue of the cross-reactivity.

Some further questions:

1) Fig. 3: which cells? Please clarify which species the cells are from and whether cell lines or primary cells. This should be clarified both in the Results section describing these experiments and in the figure legend for Fig. 3.

2) There are some instances when the word 'secretion' is used for SPMs in cell culture experiments, such as on page 1, 2, 5, 12, 13, 24, 25, 29, 33, 35, 62 and 63 in the main manuscript. The word 'presence' should be used, since the analyses of SPMs (and PGE2) were performed on the cells, neurons and microglia, i.e. not from the medium. This means that the data reflect levels in the cells, and not secreted from the cells. The wording connected to these experiments need to be corrected accordingly in the main manuscript.

The word 'secretion' is also used in Fig. 9 as a suggested mechanism, and in this case, it can remain, although it still indicates that the findings from the experiments are SPMs secreted from microglia. Indeed, the authors have shown in a previous paper data of SPMs in conditioned medium from cultured neurons.

3) Page 33: the reasoning regarding that PGE2 or other lipids were involved in the positive effect of N-AS on phagocytic activity of the microglia is not entirely convincing, i.e. there can still be lipid mediators, other than the SPMs analysed, that can contribute to the positive effects of N-AS. Also, the text in the last paragraph is incomplete, words are missing, and it is unclear. Please correct this paragraph.

4) In general, since the manuscript contains many experiments and methods, it should be easy to follow. Thus, even if the models are given in the Methods section, it is not easily followed which methods and which model system that is used when reading the Results section and reading the figure legends does not always help with the crucial information. An improvement would be to add the appropriate model and method in the Results and figure legends. It does not have to be many words, but it would help the reader to follow the studies.

Reviewer #2 (Remarks to the Author):

It is acknowledged that the authors have now performed the cross-reactivity test for the analyses of the lipid mediators. It has to be mentioned that this test indeed revealed cross-reactivity for RvD1 in the LXA4 ELISA, as well as for LXA4 in the RvD1 ELISA. These cross-reactivities are not minor, and the interpretation made in the answers by the authors is therefore not correct. However, it appears that all of the analyses of lipid mediators are now verified by LC-MS-MS analyses, and this reviewer does not object to acceptance of the manuscript on the basis of the lipid mediator analysis.

If, on the contrary, some analysis of lipid mediators is still performed by ELISA (or EIA), the authors must discuss the issue of the cross-reactivity.

We would like to thank the reviewer for these comments concerning our manuscript. Mentioned by the reviewer, although cross-reactivity for RvD1 in the LXA₄ ELISA was revealed, we have presented all results of lipid mediators using LC-MS/MS system, and we think that our results are reliable. We again thank the reviewer for these comments regarding a potential lack of clarity in our study.

Some further questions:

1) Fig. 3: which cells? Please clarify which species the cells are from and whether cell lines or primary cells. This should be clarified both in the Results section describing these experiments and in the figure legend for Fig. 3.

We would like to thank the reviewer for these comments concerning our manuscript. In figure 3, to investigate the generation and roles of N-AS in CNS cells, primary culture of neuron, microglia and astrocyte was prepared from C57BL/6 mice. This information is described in the Results and figure legend for figure 3 in the revised manuscript.

2) There are some instances when the word 'secretion' is used for SPMs in cell culture experiments, such as on page 1, 2, 5, 12, 13, 24, 25, 29, 33, 35, 62 and 63 in the main manuscript. The word 'presence' should be used, since the analyses of SPMs (and PGE₂) were performed on the cells, neurons and microglia, i.e. not from the medium. This means that the data reflect levels in the cells, and not secreted from the cells. The wording connected to these experiments need to be corrected accordingly in the main manuscript.

The word 'secretion' is also used in Fig. 9 as a suggested mechanism, and in this case, it can remain, although it still indicates that the findings from the experiments are SPMs secreted from microglia. Indeed, the authors have shown in a previous paper data of SPMs in conditioned medium from cultured neurons.

According to the reviewer's comment, we have carefully reconsidered our manuscript. We

agree with the reviewer's comment that the word 'secretion' is not appropriate for SPM in our experimental system, because we analyzed SPMs on the cells, not the media.

The treatment of N-AS increased the levels of SPMs via COX2 acetylation in neuron and microglia, indicating that N-AS induced production of SPMs. We wish to point out this function of N-AS, producing SPMs in CNS cells, and think that the word 'production' to be appropriate for describing the role of N-AS in cells rather than the word "presence" suggested by reviewer. Accordingly, we have replaced the word 'secretion' with the word 'production' on page 1, 2, 5, 12, 13, 24, 25, 29, 33, 35, 62 and 63 in the revised manuscript, and hope the reviewer understands our preference.

3) Page 33: the reasoning regarding that PGE₂ or other lipids were involved in the positive effect of N-AS on phagocytic activity of the microglia is not entirely convincing, i.e. there can still be lipid mediators, other than the SPMs analysed, that can contribute to the positive effects of N-AS. Also, the text in the last paragraph is incomplete, words are missing, and it is unclear. Please correct this paragraph.

We would like to thank the reviewer for this comment concerning our manuscript. We agree with reviewer's comment that we cannot completely rule out the possibility that the other lipid mediators, rather than the SPMs, may contribute to the positive effects of N-AS. In **Supplementary table 3**, we performed lipid mediators metabololipidomics in human microglia treated with N-AS, acetyl-CoA or zileuton in presence of A β 50 μ M or not. We confirmed that SPM precursors and SPMs were decreased in A β -treated microglia compared with control microglia, and it was restored by N-AS treatment in A β -treated microglia, showing the positive effects of N-AS-triggered SPM. Also, our results showed reduction of PGE₂/PGE₃ in A β -treated microglia compared with control microglia, and the levels of PGE₂/PGE₃ increased in presence of N-AS. Both SPMs and PGE₂, which changed by N-AS treatment in A β -treated microglia, play an important role in phagocytic ability of microglia, and phagocytosis is restored through the increase of SPMs production and decrease of PGE₂ synthesis in AD environment (*J Clin Invest.* 2015;125(1):350–364.). In our result, although the levels of PGE₂ were not decreased, the phagocytic ability was restored in A β -treated microglia in the presence of N-AS (**Supplementary Fig. 6e and f**). Therefore, we think that the positive effect of N-AS, including phagocytosis, was mediated by increase of SPM rather than PGE₂. In addition, the other lipid mediators were not altered in groups, indicating that the other lipid mediators, except SPMs and PGE₂/PGE₃, might not affect the positive effects of N-AS. Overall, these results suggest that the N-AS-triggered SPMs, rather than the other

lipid mediators, were might more responsible for the positive effects of N-AS including phagocytosis in AD environment.

Furthermore, as suggested by reviewer, we have corrected the last paragraph of page 33 in the revised manuscript. We have revised this sentence to: “These results suggest that the N-AS-triggered SPMs, rather than the other lipid mediators, were more responsible for direct regulation of microglia including phagocytosis in AD environment.”

Bioactive metabolome	Q1	Q3	LM in human microglia (ng/ml*10 ⁶ cell)						
			Control	Control + N-AS	Control + Zil + N-AS	A β	A β + Acetyl-CoA	A β + N-AS	A β + Zil + N-AS
AA bioactive metabolome									
PGH ₂ /PGE ₂	351	271	116.5±1.1	105.8±1.1	79.5±0.9****	78.7±0.8**	103.1±1.0 [#]	95.7±0.9 [#]	76.0±1.0*** [€]
TxB ₂	369	169	ND	ND	ND	ND	ND	ND	ND
11-HETE	319	167	168.5±1.8	161.8±0.9	162.5±0.2	174.9±2.0	151.2±0.5	148.4±1.9	144.4±0.9
LTB ₄	335	195	ND	ND	ND	ND	ND	ND	ND
15-HETE	319	219	647.1±7.0	625.5±6.8	610.2±4.4	564.3±4.4**	609.4±1.9 [#]	625.4±1.7 [#]	612.5±2.7 [#]
15R-LXA ₄	351	115	31.8±0.4	28.7±0.3	0.03±0.0****	0.03±0.0**	29.6±0.3 [#]	19.8±0.2 [#]	0.02±0.0*** ^{€€€}
EPA bioactive metabolome									
PGE ₃ /PGD ₃	349	269	142.9±2.2	113.1±1.6	130.0±1.9	163.9±2.5	158.7±2.5	132.9±2.0	146.7±0.6
TxB ₃	367	169	ND	ND	ND	ND	ND	ND	ND
11-HEPE	317	167	484.4±5.8	451.8±4.6	483.1±2.4	490.2±2.4	467.8±4.5	371.7±4.0	480.6±2.3
15-HEPE	317	219	298.9±5.0	284.1±4.0	273.7±4.2	272.1±7.3	297.5±3.8	237.9±3.7	268.2±2.1
18-HEPE	317	215	1729.2±24.1	1636.9±19.7	1650.0±7.6	1515.3±10.4**	1881.1±22.9 [#]	1763.6±30.4 [#]	1668.4±4.3 [#]
RvE1	349	195	22.3±0.3	18.5±0.4	0.01±0.0****	0.01±0.0***	14.5±0.2 [#]	8.6±0.3 [#]	0.01±0.0*** ^{€€€}
DHA bioactive metabolome									
13-HDHA	343	193	146.6±2.1	133.9±2.2	137.6±1.6	128.2±1.8	147.4±1.0	133.0±1.8	140.8±2.9
17-HDHA	343	281	693.3±8.5	633.8±9.2	642.3±12.4	561.8±2.7**	720.8±5.3 [#]	707.6±8.2 [#]	705.4±6.0 [#]
RvD1	375	141	2.5±0.02	2.36±0.02	0.01±0.0****	0.01±0.0***	2.75±0.04 [#]	2.13±0.02 [#]	0.01±0.0*** ^{€€€}
RvD2	375	277	ND	ND	ND	ND	ND	ND	ND
RvD3	375	147	ND	ND	ND	ND	ND	ND	ND

Human microglia were treated with N-AS, acetyl-CoA or Zil in presence of A β 50 μ M or not (n = 6 per group). All data are expressed as mean±s.e.m. One-way analysis of variance, Tukey's post hoc test. ***P* < 0.01, ****P* < 0.001 versus control. [#]*P* < 0.01, [#][#]*P* < 0.001 versus A β . **P* < 0.05, ***P* < 0.01, ****P* < 0.001 versus control+N-AS. [€]*P* < 0.05, ^{€€€}*P* < 0.001 versus A β +N-AS.

Supplementary Table 3 in the revised manuscript. Human Microglia LM-SPM profiles from Control treated with N-AS, acetyl-CoA or Zil in presence of A β 50 μ M or not.

4) In general, since the manuscript contains many experiments and methods, it should be easy to follow. Thus, even if the models are given in the Methods section, it is not easily followed which methods and which model system that is used when reading the Results section and reading the figure legends does not always help with the crucial information. An improvement would be to add the appropriate model and method in the Results and figure legends. It does not have to be many words, but it would help the reader to follow the studies.

According to the reviewer's comment, we have carefully reviewed our manuscript, and tried to provide the appropriate model and method in the Results and figure legends for the reader to understand our study.

We thank the referee for helping to make our findings more coherent and precise.